# Excitatory and inhibitory effects of HCN channel modulation on excitability of layer V pyramidal cells

**Tuomo Mäki-Marttunen** [1,2,3]\*, **Verónica Mäki-Marttunen** [4]

**1** Faculty of Medicine and Health Technology, Tampere University, Tampere, Finland, **2** Department of Biosciences, University of Oslo, Oslo, Norway, **3** Simula Research Laboratory, Oslo, Norway, **4** Cognitive Psychology Unit, Faculty of Social Sciences, University of Leiden, Leiden, Netherlands

\* tuomo.maki-marttunen@tuni.fi

**Data Availability Statement:** All relevant data are within the manuscript and its

## Abstract

Dendrites of cortical pyramidal cells are densely populated by hyperpolarization-activated cyclic nucleotide-gated (HCN) channels, a.k.a. $I_h$ channels. $I_h$ channels are targeted by multiple neuromodulatory pathways, and thus are one of the key ion-channel populations regulating the pyramidal cell activity. Previous observations and theories attribute opposing effects of the $I_h$ channels on neuronal excitability due to their mildly hyperpolarized reversal potential. These effects are difficult to measure experimentally due to the fine spatiotemporal landscape of the $I_h$ activity in the dendrites, but computational models provide an efficient tool for studying this question in a reduced but generalizable setting. In this work, we build upon existing biophysically detailed models of thick-tufted layer V pyramidal cells and model the effects of over- and under-expression of $I_h$ channels as well as their neuromodulation. We show that $I_h$ channels facilitate the action potentials of layer V pyramidal cells in response to proximal dendritic stimulus while they hinder the action potentials in response to distal dendritic stimulus at the apical dendrite. We also show that the inhibitory action of the $I_h$ channels in layer V pyramidal cells is due to the interactions between $I_h$ channels and a hot zone of low voltage-activated $Ca^{2+}$ channels at the apical dendrite. Our simulations suggest that a combination of $I_h$-enhancing neuromodulation at the proximal part of the apical dendrite and $I_h$-inhibiting modulation at the distal part of the apical dendrite can increase the layer V pyramidal excitability more than either of the two alone. Our analyses uncover the effects of $I_h$-channel neuromodulation of layer V pyramidal cells at a single-cell level and shed light on how these neurons integrate information and enable higher-order functions of the brain.

## Author summary

Neurons undergo many types of neuromodulation that regulate the neuron excitability by enhancing or hindering the activity of different ion channels. One of the ion-channel classes that are strongly expressed in excitatory cortical neurons and strongly affected by neuromodulators such as dopamine or noradrenaline are the hyperpolarization-activated

Supporting information files. Our simulation scripts (interfaced through Python, versions 3.7.5 and 3.9.1 tested) are publicly available at http://modeldb.yale.edu/267293.

**Funding:** TMM was supported by the Academy of Finland (330776,336376), the Research Council of Norway (248828), and the University of Oslo Convergence Environment (4MENT). The funders had no role in study design, data collection and analysis, decision to publish, or preparation of the manuscript.

**Competing interests:** The authors have declared that no competing interests exist.

cyclic nucleotide-gated (HCN) channels, a.k.a. $I_h$ channels. In this work, we use computational modelling to analyze the effects that the $I_h$ channels have on deep-layer cortical neurons by simulating the blockade or various types of neuromodulation of these channels. We show that $I_h$ channels enhance the neuron activity when the neuron is stimulated at proximal dendrites and inhibit the neuron activity when the neuron is stimulated at distal dendrites. We also show that the inhibitory actions of $I_h$ channels are dependent on $Ca^{2+}$ channels. Our analyses help to understand the effects of $I_h$-channel neuromodulation of deep-layer cortical excitatory neurons and shed light on how these neurons contribute to information processing and enable cognitive functions of the brain.

## 1 Introduction

In the brain, higher-order cognition and consciousness are believed to rely on the highly specialized neurons that populate the cortex, the layer V pyramidal cells (L5PCs). Thanks to their complex morphology and connectivity, feed-forward sensory-related stimuli and feed-back context-dependent inputs arrive at spatially distinct sections of the L5PC dendritic tree, the former stimulating the basal dendrites and the latter largely exerting their action on the distal apical dendrite [1]. The different inputs are integrated in the soma and together determine the specific patterns of activity of the neuron. The effect that feedback context-dependent inputs have on somatic excitability partly depends on hyperpolarization-activated cyclic nucleotide-gated (HCN) channels, a.k.a. $I_h$ channels. L5PCs strongly express $I_h$ channels in their apical dendrite that reaches up to layer I of the cortex [2]. Moreover, the apical dendrite of L5PCs is abundantly projected to by neuromodulatory terminals from subcortical regions, including ventral tegmental area (VTA, dopaminergic neuromodulation) [3], basal forebrain (cholinergic modulation) [4], and locus coeruleus (noradrenergic modulation) [5]. The interplay between neuromodulatory input and $I_h$-driven communication between apical and somatic sections confers L5PC neurons their complex integrative capacity. However, the details of the mechanisms behind this interplay remain an open question.

$I_h$ channels give the neuron an extensive set of modes of excitability. A reason for this is that they can either depolarize or hyperpolarize the cell membrane during subthreshold membrane potential fluctuation. That is, their reversal potential lies between -45 and -30 mV [6, 7] and therefore their effect on membrane excitability will depend on the electrical environment. In addition, the $I_h$ channels can be modulated by several neuromodulators such as dopamine, acetylcholine and norepinepherine [8–10]. Previous experimental work has assessed separately the effects of $I_h$ blockage or neuromodulation in terms of their effect on somatic [11] and apical dendritic [12, 13] excitability. However, the exact way in which the concurrence of the multiple factors affect the direction of $I_h$ modulation on neuron excitability remains unclear. Computational modelling offers the possibility to assess the mechanisms behind cellular electrical properties, and generate testable predictions. In this work, we used biophysically detailed computational modelling to analyze the effect of $I_h$ channels and their neuromodulation on L5PC activity.

To analyze the effects of $I_h$ channels and the way they modulate L5PC excitability, we used existing biophysically detailed neuron models of thick-tufted L5PCs with reconstructed morphologies. Unlike thin-tufted L5PCs that project across hemispheres through the corpus callosum, thick-tufted L5PCs express strong $I_h$ currents and mostly project to subcortical structures [14, 15]. We determined the threshold currents or conductances for many types of stimulus protocols in presence and absence of $I_h$ currents. In this way, we characterized the types and

**Table 1. Neuromodulators acting on $I_h$ channels, mediated by cAMP.** *Note that not all of these neuromodulatory pathways may take place in mammalian L5PCs, and some interactions depend on the age and species (see Discussion).

| Neuromodulator | Receptor | Effect on $I_h$*, references |
|---|---|---|
| Dopamine | D1 | Enhancing [16, 17] |
| Noradrenaline | $\beta$ | Enhancing [18] |
| Serotonin | 5-HT7 | Enhancing [19, 20] |
| Dopamine | D2 | Inhibiting [21] |
| Noradrenaline | $\alpha 2$ | Inhibiting [22] |
| Acetylcholine | M2 | Inhibiting [23] |
| Serotonin | 5-HT1 | Inhibiting [24] |

locations of stimuli for which $I_h$ currents facilitate action potential (AP) initiation (i.e., are excitatory) and those for which they are shunting (i.e., inhibitory). We also modelled the response of the neuron when the $I_h$ channels were under neuromodulation. $I_h$ channels are bound to neuromodulatory effects through the cAMP intracellular pathway. cAMP binds to an $I_h$ channel and increases its open probability. We examined the effect of cAMP-enhancing or cAMP-inhibiting neuromodulators (Table 1) by introducing experimentally observed effects of these neuromodulators on the voltage-dependence profile of $I_h$ channels. We showed that $I_h$ channels shunt stimulation at the distal apical dendrite of L5PCs but facilitate the AP induction for proximal inputs. By using two models with different $Ca^{2+}$ channel distributions we showed that the shunt-inhibitory effect of the $I_h$ channels requires presence of a hot zone of low-voltage activated (LVA) $Ca^{2+}$ channels at the mid-distal apical dendrite. Furthermore, we showed that neuromodulators had a similar bimodal location-dependent effect on L5PC excitability. We also demonstrated that maximal neuromodulatory effects can be brought about by combining $I_h$-enhancing neuromodulation at the proximal dendrite and $I_h$-inhibiting neuromodulation at the distal dendrite, or vice versa. Our analysis uncovers the effects that neuromodulation of $I_h$ can have on L5PCs at a single-cell level. This sheds new light on how L5PCs integrate information and enable higher-order functions of the neocortex.

## 2 Methods

### 2.1 Neuron models

We employed two models of L5PCs: the "Hay model" [25] and the "Almog model" [26]. Both models were multicompartmental Hodgkin-Huxley type of models with reconstructed layer V thick-tufted pyramidal neuron morphologies. The ionic current species of the two models are listed in Table 2. In the Hay model, the ion-channel conductances were constant along the dendrites, except for the $I_h$ channel whose conductance grows exponentially with the distance from the soma and the $Ca^{2+}$ channels where a hot zone of Conductance of apical dendritic $I_h$ channels grew exponentially with the distance from the soma. For the $Ca^{2+}$ channels, a hot zone ($10\times$ larger HVA $Ca^{2+}$ channel conductance and $100\times$ larger LVA $Ca^{2+}$ channel conductance) was present at the apical dendrite at a distance from 685 to 885 μm from the soma [25, 27]. In the Almog model, there was no hot zone of $Ca^{2+}$ channels, but all ion-channel conductances varied spatially (usually piece-wise linear) along the dendrites [26]. The distribution of the $I_h$ channels in the two models are illustrated in S1 Fig. In addition to describing the dynamics of these ionic channels, the models also describe the dynamics of the intracellular $Ca^{2+}$ concentration, $[Ca^{2+}]_i$, which affects the currents conducted by SK and BK channels. According to the models, $[Ca^{2+}]_i$ is increased by the current flow through $Ca^{2+}$ channels, and is otherwise decreased towards a resting-state level of $[Ca^{2+}]_i$.

**Table 2. Current species in the Hay and Almog models.**

| Hay Model | Almog model |
|---|---|
| Fast inactivating Na$^+$ current ($I_{Nat}$) | Fast inactivating Na$^+$ current ($I_{Nat}$) |
| Non-specific cation current ($I_h$) | Non-specific cation current ($I_h$) |
| High-voltage-activated Ca$^{2+}$ current ($I_{CaHVA}$) | High-voltage-activated Ca$^{2+}$ current ($I_{CaHVA}$) |
| Low-voltage-activated Ca$^{2+}$ current ($I_{CaLVA}$) | Medium-voltage-activated Ca$^{2+}$ current ($I_{CaMVA}$) |
| Fast inactivating K$^+$ current ($I_{Kt}$) | Fast inactivating K$^+$ current ($I_{Kt}$) |
| Slow inactivating K$^+$ current ($I_{Kp}$) | Slow inactivating K$^+$ current ($I_{Kp}$) |
| Small-conductance Ca$^{2+}$-activated K$^+$ current ($I_{SK}$) | Small-conductance Ca$^{2+}$-activated K$^+$ current ($I_{SK}$) |
| Fast non-inactivating K$^+$ current ($I_{Kv3.1}$) | Large-conductance voltage and Ca$^{2+}$-gated K$^+$ current ($I_{BK}$) |
| Muscarinic K$^+$ current ($I_m$) | Passive leak current ($I_{leak}$) |
| Persistent Na$^+$ current ($I_{Nap}$) | |
| Passive leak current ($I_{leak}$) | |

The dynamics of the $I_h$ current, gated by the inactivation variable $h$ as $I_h = \bar{g}_h h (E_h - V_m)$, is described as follows. In both models, the variable $h$ obeys the equation

$$\frac{dh}{dt} = \frac{1}{\tau_\infty}(h_\infty - h)$$

In the Hay model, $h_\infty$ and $\tau_\infty$ are described as follows [25, 28]:

$$\alpha_h = -\frac{v_{\text{off},a} - V_m}{\tau_a}\frac{1}{\exp(-(v_{\text{off},a} - V_m)/v_{\text{slo}}) - 1}$$

$$\beta_h = \frac{\exp(-(v_{\text{off},b} - V_m)/v_{\text{slo},b})}{\tau_b} \tag{1}$$

$$h_\infty = \frac{\alpha_h}{\alpha_h + \beta_h}$$

$$\tau_\infty = \frac{1}{\alpha_h + \beta_h}$$

with $v_{\text{off},a}$ = -154.9 mV, $v_{\text{slo},a}$ = 11.9 mV, $\tau_a$ = 155.521 ms, $v_{\text{off},b}$ = 0.0 mV, $v_{\text{slo},b}$ = 33.1 mV, and $\tau_b$ = 5.18135 ms. In the Almog model, $h_\infty$ and $\tau_\infty$ are described as follows [26, 29]:

$$h_\infty = \frac{1}{1 + \exp((V_m - v_{\text{off}})/v_{\text{slo}})} \tag{2}$$

$$\tau_\infty = \frac{1}{t_{adj}\left(\frac{1}{t_0}\exp((v_{\text{off},t1} - V_m)/v_{\text{slo},t1}) + \frac{1}{t_1}\exp(-(v_{\text{off},t2} - V_m)/v_{\text{slo},t2})\right)}$$

$$t_{adj} = q_{10}^{(T-22^\circ \text{C})/10^\circ \text{C}}$$

where $v_{\text{off}}$ = -91 mV, $v_{\text{slo}}$ = 6 mV, $t_0$ = 2542.5883549 ms, $t_1$ = 11.40250855 ms, $v_{\text{off},t1}$ = 0 mV, $v_{\text{off},t2}$ = 0 mV, $v_{\text{slo},t1}$ = 40.1606426 mV, $v_{\text{slo},t2}$ = 16.1290323 mV, $q_{10}$ = 2.3, and $T$ = 34°C. The reversal potential of the $I_h$ current was more depolarized in the Almog model ($E_h$ = -33 mV) than in the Hay model ($E_h$ = -45 mV).

The dynamics of the LVA $Ca^{2+}$ current in the Hay model, gated by activation and inactivation variables $m$ and $h$ as $I_{CaLVA} = \bar{g}_{CaLVA} m^2 h (E_{Ca^{2+}} - V_m)$, is described as follows:

$$\frac{dm}{dt} = \frac{1}{\tau_{m,\infty}} (m_\infty - m) \tag{3}$$

$$\frac{dh}{dt} = \frac{1}{\tau_{h,\infty}} (h_\infty - h) \tag{4}$$

$$m_\infty = \frac{1}{1 + \exp((v_{off,m} - V_m)/v_{slo,m})} \tag{5}$$

$$\tau_m = \frac{1}{t_{adj}} \left( \tau_{m,min} + \frac{\tau_{m,diff}}{1 + \exp(-(v_{off,m,t} - V_m)/v_{slo,m,t})} \right)$$

$$h_\infty = \frac{1}{1 + \exp((v_{off,h} - V_m)/v_{slo,h})} \tag{6}$$

$$\tau_h = \frac{1}{t_{adj}} \left( \tau_{h,min} + \frac{\tau_{h,diff}}{1 + \exp(-(v_{off,h,t} - V_m)/v_{slo,h,t})} \right)$$

where $v_{off,m} = $ -40.0 mV, $v_{off,m,t} = $ -35.0 mV, $v_{off,h} = $ -90.0 mV, $v_{off,h,t} = $ -50.0 mV, $v_{slo,m} = 6.0$ mV, $v_{slo,m,t} = 5.0$ mV, $v_{slo,h} = 6.4$ mV, $v_{slo,h,t} = 7.0$ mV, $\tau_{m,min} = 5.0$ ms, $\tau_{m,diff} = 20.0$ ms, $\tau_{h,min} = 20.0$ ms, and $\tau_{h,diff} = 50.0$ ms. The reversal potential is dependent on the local intracellular $Ca^{2+}$ concentration and is thus a dynamic variable—in our simulations it was typically around 100–130 mV but could reach values as low as 40 mV in the Almog model (S12(A) and S12(B) Fig). For the description of the other current species, we refer to the original publications [25, 26].

All simulations were run using NEURON software (version 7.8.2) with adaptive time-step integration. Threshold currents and conductances were sought for using bisection method. Our simulation scripts (interfaced through Python, versions 3.7.5 and 3.9.1 tested) are publicly available at http://modeldb.yale.edu/267293.

## 2.2 Stimulation protocols

In Section 3.1, we stimulated the center of the soma with a square pulse current, starting at 200 ms and lasting until the end of the simulation (16 s). The spiking frequency was determined based on the number of spikes from 500 to 16000 ms. In Section 3.2, we stimulated dendritic sections with a short square pulse current (0.2 ms) or with a conductance-based, alpha-shaped (time constant 5 ms) glutamatergic (reversal potential $E_{glu} = 0$ mV) input. When choosing the location along the dendrite the thickest dendritic section at a given distance was selected as in [25]. In Sections 3.3–3.4, the glutamatergic synaptic inputs were modelled with more precision (except for the simulations of Fig 5D and 5E), similar to [30, 31]: The AMPAR-mediated currents were modelled as

$$I_{AMPAR} = g_{AMPAR} s_{AMPAR} (E_{glu} - V_m),$$

where the synaptic variable $s$ increased instantaneously with incoming spikes and decayed exponentially as $\frac{ds_{AMPAR}}{dt} = -\frac{1}{\tau_{s,AMPAR}} s_{AMPAR} + \delta(t - t_{spike})$ with a time constant $\tau_{s,AMPAR} = 2$ ms.

The NMDAR-mediated currents were modelled as

$$I_{\mathrm{NMDAR}} = g_{\mathrm{NMDAR}} s_{\mathrm{NMDAR}} \frac{1}{1 + [\mathrm{Mg}^{2+}]\exp(-0.062/\mathrm{mV}/V_m)/3.57\mathrm{mM}} (E_{\mathrm{glu}} - V_m),$$

where the synaptic variables $s_{\mathrm{NMDAR}}$ and $x_{\mathrm{NMDAR}}$ obeyed the following dynamics:

$$\frac{ds_{\mathrm{NMDAR}}}{dt} = -\frac{1}{\tau_{s,\mathrm{NMDAR}}} s_{\mathrm{NMDAR}} + \alpha_s x_{\mathrm{NMDAR}}(1 - s_{\mathrm{NMDAR}}) \ ,$$

$$\frac{dx_{\mathrm{NMDAR}}}{dt} = -\frac{1}{\tau_{x_{\mathrm{NMDAR}}}} x_{\mathrm{NMDAR}} + \delta(t - t_{\mathrm{spike}}).$$

The rise time was $\tau_{x,\mathrm{NMDAR}} = 2$ ms and the decay time $\tau_{s,\mathrm{NMDAR}} = 100$ ms, and the rate of current activation was set $\alpha_s = 2$ kHz as in [30]. The NMDAR-mediated current was also used Fig 3E in Section 3.2, where $g_{\mathrm{AMPAR}} = 0$ uS, whereas in Sections 3.3–3.4 the conductances $g_{\mathrm{AMPAR}}$ and $g_{\mathrm{NMDAR}}$ were set the same. In simulations containing the AMPA- and NMDA-receptor or GABA-receptor synapses, we distributed 2000 simultaneously activated synapses across the apical or basal dendrite, typically restricting to distances $[x_1, x_2]$ from the soma, where the parameters $x_1$ (x-axis) and $x_2$ (y-axis) ranged from 200 μm to 1300 μm in intervals or 100 μm. In Section 3.4, we also modelled the effects of concurrent GABAergic inhibition of the basal dendrite, which was modelled the same way as AMPAR-mediated currents but the reversal potential was set -80 mV. The spike time $t_{\mathrm{spike}}$ of the AMPAR- and NMDAR-mediated inputs was the same (2000 ms) for all synapses, while the spike time of GABAergic inputs was randomly picked between 1975–2025 ms.

## 2.3 Alterations of ion-channel properties

**Blockage of $I_h$.** We blocked the $I_h$ currents by setting the maximal conductance to 0 everywhere (unless otherwise stated) in the neuron.

**cAMP-enhancing and cAMP-inhibiting neuromodulation of $I_h$.** We modelled the cAMP-enhancing modulation of $I_h$ by increasing the half-inactivation voltage by +10 mV or +5 mV in the Hay or Almog model, respectively, and the cAMP-inhibiting modulation of $I_h$ by decreasing it by the same amount. See Table 1 for the neuromodulator receptors mediating these effects. We used the smaller voltage difference (±5 mV) for the Almog model due to the relatively large effect of ±10 mV shifts (S12(C) Fig). To model weakly cAMP-dependent $I_h$ channels (HCN1 homomeric channels) in S7–S8 Figs, we used smaller half-inactivation voltage shifts, namely ±4 mV and ±2 mV in Hay and Almog models, respectively.

**Removal of the hot zone of $Ca^{2+}$ channels from the Hay model.** We used the same conductance of HVA and LVA $Ca^{2+}$ channels in the area of the hot zone as elsewhere in the apical dendrite ($g_{\mathrm{CaHVA}} = 55.5$ uS/cm$^2$, $g_{\mathrm{CaLVA}} = 187$ uS/cm$^2$).

**Addition of a hot zone of $Ca^{2+}$ channels to the Almog model.** We adopted the LVA $Ca^{2+}$ current from the Hay model in addition to the HVA and MVA $Ca^{2+}$ currents native to the Almog model. We added LVA $Ca^{2+}$ channels to the apical dendrite with a maximal conductance of 300 mS/cm$^2$ for dendritic sections with a distance of 585–985 μm to the soma and 3 mS/cm$^2$ for the others.

# 3 Results

## 3.1 Activation of $I_h$ increases L5PC activity when stimulated at soma

$I_h$ blockage has been shown, both experimentally (e.g., by application of ZD7288 [32, 33]) and computationally [34], to lead to decreased neuron spiking. Here, we replicated this result with

our biophysically detailed neuron models of a L5PC neuron, namely, the Almog model (Fig 1A) [26] and the Hay model (Fig 1B) [25], by considering the f-I curve of the L5PC under various operations on the $I_h$ channel. See S1 Fig for the $I_h$ channel distribution in the two models. In both Almog (Fig 1C) and Hay (Fig 1D) models, the $I_h$ blockage made the resting membrane potential more hyperpolarized. Moreover, a block of $I_h$ decreased the rate of firing in response to DC applied to soma, both in the Almog and Hay models, although the effects were considerably larger in the Almog model (Fig 1E, blue) than in the Hay model (Fig 1F, blue). On the other hand, an increase in $I_h$ conductances increased the firing rates in both models (Fig 1E and 1F, red curves). Likewise, a model of neuron-wide cAMP-inhibiting modulation (half-inactivation voltage decreased) decreased the firing rates, and a cAMP-enhancing modulation (half-inactivation voltage augmented) increased the firing rates in both models (Fig 1G and 1H). This was also the case when $I_h$ currents were blocked or enhanced only at the dendrites (S2(A)–S2(D) Fig) instead of both soma and dendrites: both $I_h$ blockage (S2(A) and S2(B) Fig, blue) and cAMP-inhibiting modulation (S2(C) and S2(D) Fig, blue) at the dendrites decreased the L5PC firing rates, while $I_h$ over-expression (S2(A) and S2(B) Fig, red) and cAMP-enhancing modulation (S2(C) and S2(D) Fig, red) at the dendrites increased the firing rates. The -10 mV shift applied in the Hay model as a model of cAMP-inhibiting neuromodulation was in agreement with the D2-mediated effects of 500 nM dopamine observed in [3]: our model reproduced the numbers of spikes (7 in control case, 5 under dopaminergic modulation, see Figure 6 in [3]) in response to 1-second stimulation (S2(E) and S2(F) Fig). Taken together, these results support the role of $I_h$ current as an enhancer of L5PC activity when the neuron is stimulated at the soma.

## 3.2 $I_h$ activation can increase AP threshold in response to apical dendritic stimulation in an L5PC when a hot zone of Ca$^{2+}$ channels is present

The reversal potential of the $I_h$ typically lies in the range -45–-30 mV, which grants the $I_h$ channel the possibility to shunt, that is, to inhibit inputs that would otherwise depolarize the membrane to potentials higher than this. Indeed, in [35], increased $I_h$ activation by lamotrigine (an enhancer of $I_h$ channels) application led to decreased firing response to dendritic stimuli in a CA1 neuron, and they reproduced their findings with a computational model consisting of a realistic morphology and one type of sodium and two types of potassium channels. In line with this, in [36], pharmacological blockage of $I_h$ increased the amplitude of distally elicited EPSPs compared to proximally elicited ones in a CA1 neuron. Computational models have suggested that the observed phenomena could be caused by secondary mechanisms, where a blockage or altered expression of $I_h$ channels also indirectly affects conductance of other ion channels, such as Twik-related acid-sensitive K$^+$ (TASK) channels [37, 38]. Here, we explored the possibility that the inhibitory actions attributed to $I_h$ activity in L5PCs are caused by direct shunting effects, without concurrent changes in the conductance of other ion channels.

We simulated the injection of a strong dendritic square-pulse current stimulus of 0.2 ms duration that locally depolarizes the dendrite. We measured the somatic response in the presence of $I_h$ current and compared it to the response in absence or partial absence of the current. We varied the site of dendritic stimulation from 50 to 1000 μm with an interval of 50 μm and used the bisection method to find the AP threshold for each stimulation site.

The Almog model neuron consistently predicted that the presence of $I_h$ current facilitated the AP initiation by a dendritic stimulus (Fig 2A–2D). Blocking the $I_h$ currents hyperpolarized the basal membrane potential and increased the threshold current for inducing a spike with a dendritic stimulus at a distance of 500 μm from the soma (Fig 2A). The threshold current was increased in the Almog model across the apical dendrite (Fig 2B). For stimulation sites further than 400–500 μm from the soma the threshold currents implied unrealistically high (> 100

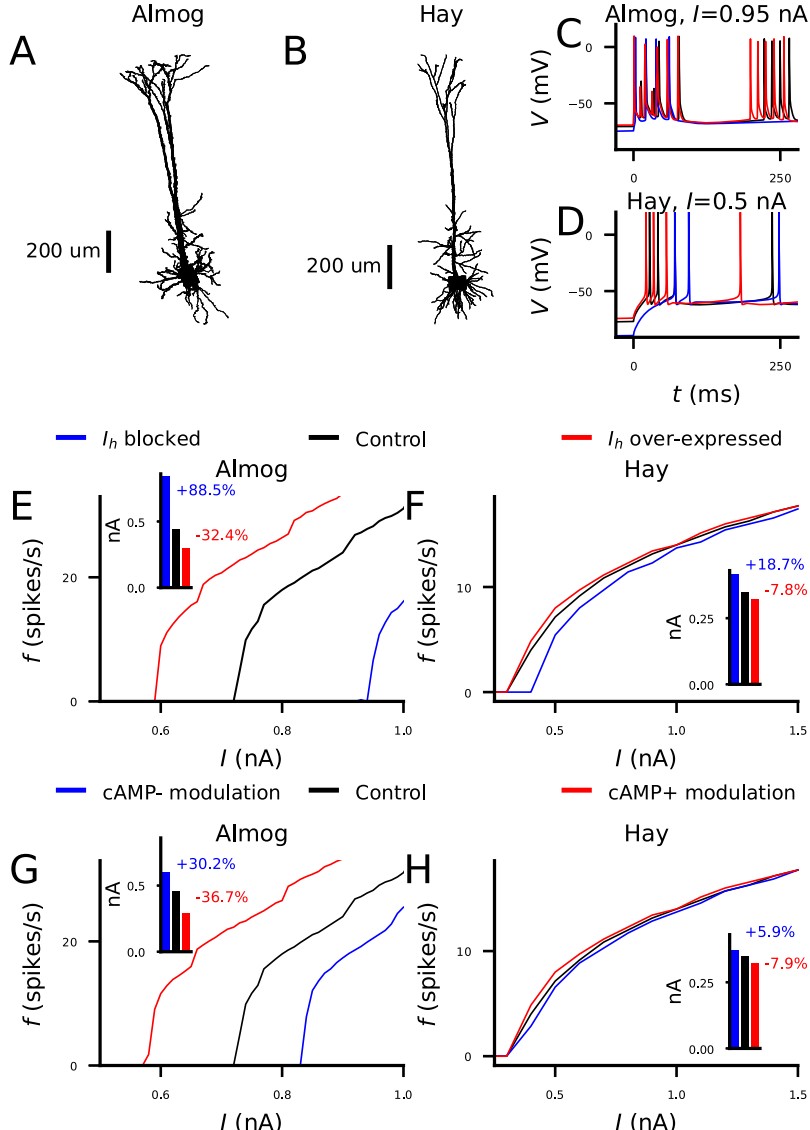

**Fig 1.** $I_h$ **activation increases the frequency of action potentials in response to somatic DC in L5PCs. A–B**: The morphology of the Almog (A) and Hay (B) model neurons. **C–D**: Membrane potential time courses of the Almog (C) and Hay (D) model neurons. **E–F**: The frequency of APs (y-axis) in response to somatic DC of a given amplitude (x-axis) in Almog (E) and Hay (F) model neurons under up- or down-regulated $I_h$ channels. Black: control neuron. Blue: $I_h$ conductance blocked. Red: $I_h$ conductance increased by 100%. **G–H**: The frequency of APs response to somatic DC in Almog (G) and Hay (H) model neurons under different neuromodulatory states. Black: control neuron. Blue: cAMP-inhibiting neuromodulation, modelled as a -5 mV (G) or -10 mV (H) shift in half-inactivation potential of the $I_h$ channels. Red: cAMP-enhancing neuromodulation, modelled as a +5 mV (G) or +10 mV (H) shift in half-inactivation potential of the $I_h$ channels. The insets of panels (E)–(H) show the threshold current amplitudes for inducing an AP. The relative changes to the threshold current amplitude of the control model are displayed next to the corresponding bars in the insets (the threshold current amplitudes can be different from the onsets of the f-I curves since some DC amplitudes only cause one spike). Note the difference in axes scales between the models in (E–H). Areas under curve, Almog model: 0.76 ($I_h$ blocked), 3.1 ($I_h$ under cAMP-inhibiting neuromodulation), 6.0 (control), 11.9 ($I_h$ overexpressed), and 13.1 ($I_h$ under cAMP-enhancing neuromodulation) nA·Hz. Areas under curve, Hay model: 14.0 ($I_h$ blocked), 14.8 ($I_h$ under cAMP-inhibiting neuromodulation), 15.2 (control), 15.6 ($I_h$ overexpressed), 15.6 ($I_h$ under cAMP-enhancing neuromodulation) nA·Hz.

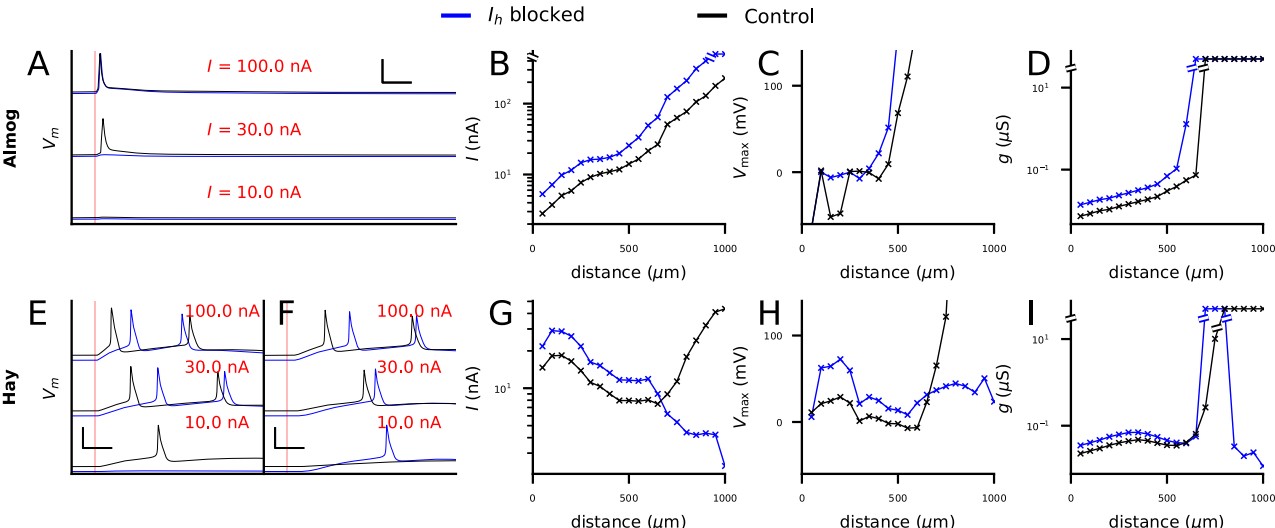

**Fig 2. $I_h$ activation may increase or decrease the threshold for L5PC action potential firing by a strong apical dendritic input, depending on the location of the input. A**: Somatic membrane potential time courses according to the Almog model in response to 2-ms square-pulse current with an amplitude of 100 (top), 30 (middle) or 10 nA (bottom), injected at the apical dendrite 500 μm from the soma. For control neuron (black), both 30 and 100 nA stimuli induced a spike, while for $I_h$-blocked neuron (blue), 100 nA stimulus induced a spike while 30 nA stimulus did not. Scale bars 5 ms and 50 mV. **B**: Threshold current amplitudes for 2-ms square-pulse inputs at the apical dendrite at different distances from the soma. Black: Almog-model control neuron, blue: Almog-model neuron with $I_h$ blockage. **C**: Peak membrane potential at the site of current injections, given the threshold-current of (B). **D**: Threshold conductances for an alpha-shaped glutamatergic input with time constant 3 ms. Black: Almog-model control neuron, blue: Almog-model neuron with $I_h$ blockage. **E–F**: Somatic membrane potential time courses according to the Hay model in response to 2-ms square-pulse current with an amplitude of 100 (top), 30 (middle) or 10 nA (bottom), injected at the apical dendrite 500 (E) or 800 (F) μm from the soma. For a stimulus 500 μm from the soma (E), 10.0 nA stimulus induced a spike in the control neuron (black) but not in the $I_h$-blocked neuron (blue), while at 800 μm this was reversed. Scale bars 5 ms and 50 mV. **G–I**: The experiments of (B–D) repeated for Hay model.

mV) membrane potentials at the site of stimulation (Fig 2C). We thus confirmed our results using conductance-based inputs: whenever a spike could be initiated by an alpha-shaped glutamatergic synaptic conductance, the threshold conductance was increased in the $I_h$-blocked model compared to the control Almog model (Fig 2D).

In contrast to the Almog model predictions, the Hay model predicted that blockage of $I_h$ current may either facilitate or hinder the AP initiation by dendritic stimulation, depending on the distance from the soma (Fig 2E–2I). Although the blockage of $I_h$ currents resulted in hyperpolarization of the baseline membrane potential at a distance of both 500 and 800 μm from the soma (Fig 2E and 2F), the threshold current was increased for stimulus at 500 μm (Fig 2E) and decreased at 800 μm (Fig 2F) from the soma. Systematically calculating the threshold currents for stimulus distances 50–1000 μm revealed that this switch of $I_h$ channels changing from excitatory to inhibitory occurred around 650–700 μm from the soma (Fig 2G). The threshold currents applied to dendritic sites further than 700 μm from the soma in the presence of $I_h$ currents caused unrealistically high local membrane potentials (Fig 2H). We thus replicated the result of Fig 2G using conductance-based stimuli (Fig 2I): the threshold conductance was larger for the $I_h$-blocked than the default Hay model when the stimulus site was closer than 800 μm from the soma, and vice versa for stimuli further than 800 μm (Fig 2I). Notably, the distance at which $I_h$ became inhibitory (650–850 μm, Fig 2G and 2I) coincides with the hot zone of $Ca^{2+}$ channels in the Hay model (685–885 um) [25].

We next analyzed the contributions of the $Ca^{2+}$ channels to the switch in threshold currents between control and $I_h$-blocked neuron models. Complete blockage of LVA $Ca^{2+}$ channels radically increased the threshold currents in the $I_h$-blocked Hay-model neuron, abolishing the switch in threshold currents (Fig 3A). Blocking any of the other voltage-gated ion channels

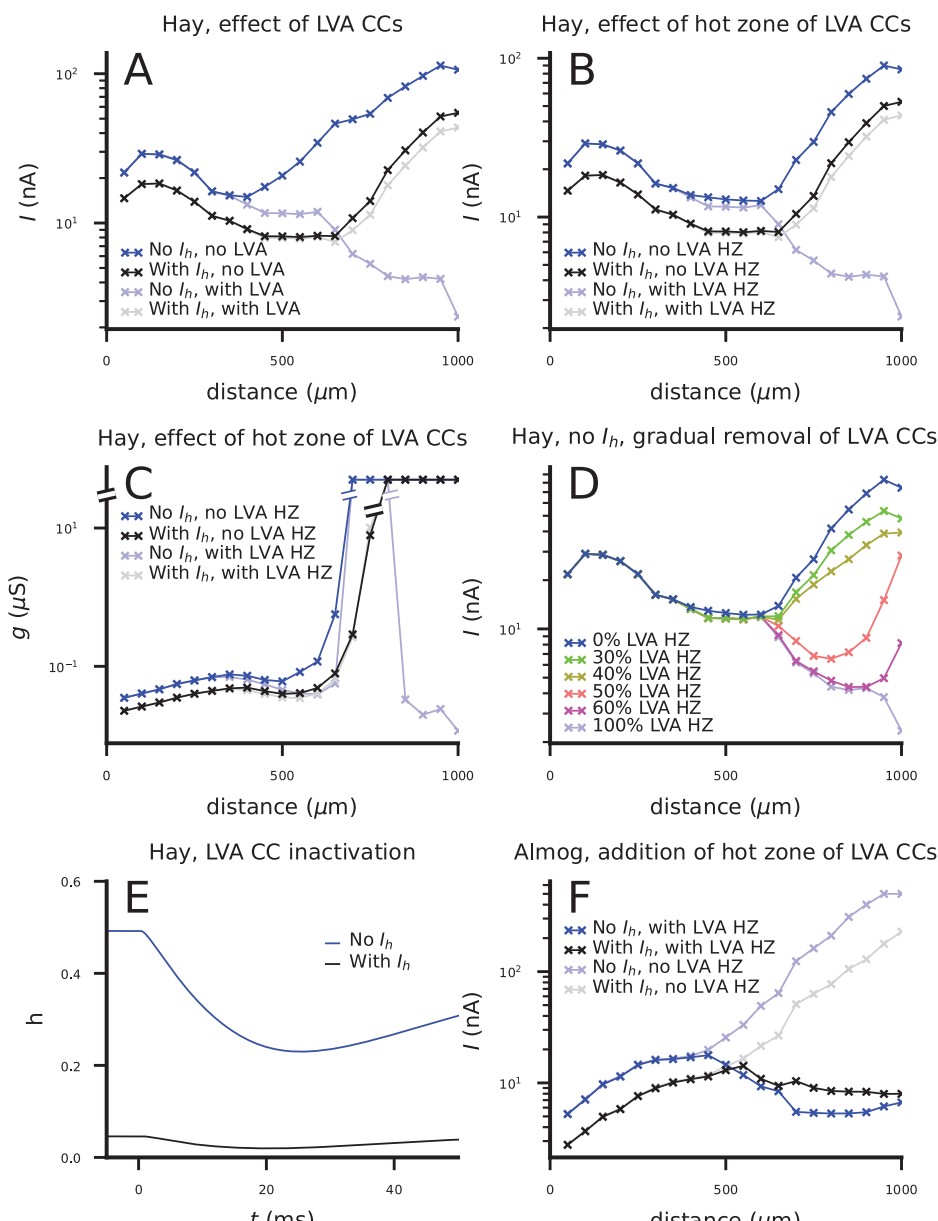

**Fig 3. The shunting inhibitory effect of $I_h$ channels is mediated by Ca$^{2+}$ channels in the apical dendrite. A**: Threshold current amplitudes for 2-ms square-pulse inputs at the apical dendrite according to Hay model without LVA Ca$^{2+}$ channels. Black: Hay-model neuron with LVA Ca$^{2+}$ channel blockage, blue: Hay-model neuron with LVA Ca$^{2+}$ and $I_h$ channel blockage. Dim curves: the data from Fig 2G where LVA Ca$^{2+}$ channels were intact. **B**: Threshold current amplitudes according to Hay model without a hot zone of LVA Ca$^{2+}$ channels with (black) or without (blue) $I_h$ channels. Dim curves: the data from Fig 2G where LVA Ca$^{2+}$ channels had higher conductance in the hot zone (650–850 μm from the soma). **C**: Threshold conductances for an alpha-shaped inputs from Fig 2D in a Hay-model neuron without a hot zone of LVA Ca$^{2+}$ channels with (black) or without (blue) $I_h$ channels. Dim curves: the corresponding data from a Hay model where LVA Ca$^{2+}$ channels were intact. **D**: Threshold current amplitudes according to Hay model without $I_h$ currents where the conductance of the LVA Ca$^{2+}$ channels in the hot zone was gradually reduced toward the baseline LVA Ca$^{2+}$ channel conductance in the apical dendrite (0% and 100% curves are identical to the corresponding curves in (B)). **E**: The time course of the inactivation variable $h$ of $I_h$ according to the Hay model (see Eq 4) in response to an alpha-shaped conductance (onset at t = 0 ms) of amplitude 0.01 nS. **F**: Threshold current amplitudes in an Almog-model neuron where MVA Ca$^{2+}$ channels in the apical dendrite were replaced by LVA Ca$^{2+}$ channels with conductance 0.003 S/cm$^2$, except for apical dendritic sections at a distance of 585–985 μm from soma where the conductance was 0.3 S/cm$^2$, with (black) or without (blue) $I_h$ channels. Dim curves: the data from Fig 2B where the Ca$^{2+}$ channels were as in the default Almog model.

(see Table 2) from the Hay model did not change the qualitative behaviour of Fig 2G (S3 Fig). The relatively small effects of blockade of HVA $Ca^{2+}$ channels and SK channels was surprising in light of our previous computational studies highlighting the role of these channels in shaping L5PC activity [39, 40]. In fact, it was sufficient to only remove the excessive LVA $Ca^{2+}$ channels from the hot zone of $Ca^{2+}$ channels: when the same LVA $Ca^{2+}$ channel conductances were used in the hot zone as in the rest of the apical dendrite while other ion-channel conductances were untouched, $I_h$ current blockage always led to an increased threshold current, similar to the Almog model (Fig 3B). We also replicated this result with conductance-based inputs (Fig 3C). We confirmed this result by decreasing the LVA $Ca^{2+}$ channel conductance at the hot zone little by little: the shunting effect of $I_h$ current disappeared in the Hay model when the LVA $Ca^{2+}$ channel conductance was reduced to approximately 30–40% from that in the control Hay model (Fig 3D). As suggested by previous modelling work of CA1 neurons [12], the reason behind the interaction of LVA $Ca^{2+}$ currents and the $I_h$ current was that in the presence of $I_h$ currents, the LVA $Ca^{2+}$ channels were highly inactivated (see Eqs 4 and 6) at resting membrane potential (Fig 3E). Namely, the inactivation variable $h$ of the LVA currents measured 800 μm from the soma had a resting value of 0.05 in the presence of $I_h$ currents and 0.49 in the absence of $I_h$ currents (Fig 3E), suggesting many times stronger resting-state LVA currents in the $I_h$ blocked case and thus strongly facilitated $Ca^{2+}$ spike generation compared to the control case. We confirmed the decisive role of the LVA $Ca^{2+}$ channel inactivation with simulations of an isolated Hay-model compartment from the distal apical dendrite: when the time course of activation variable $m$ of the LVA $Ca^{2+}$ channels (Eq 3) was artificially replaced by the corresponding time course recorded in the absence of $I_h$ channels, the $I_h$-blocked dendrite remained more excitable than the dendrite with $I_h$ channels intact, but when the time course of inactivation variable $h$ (Eq 4) was replaced by that recorded in the absence of $I_h$ channels, the shunting effect disappeared (S4 Fig). In line with these observations, when a hot zone of LVA $Ca^{2+}$ channels was added to the Almog model, we observed a qualitatively similar switch of threshold current amplitudes between $I_h$-blocked and control case (Fig 3F). The Almog model implemented with the hot zone of LVA $Ca^{2+}$ channels produced dendritic $Ca^{2+}$ spikes that typically induced a burst of APs (S5 Fig), similar to the Hay model [25]. Taken together, our simulations suggest $I_h$ channel activity can increase the threshold current in distal dendritic stimuli (i.e., it can have a shunting inhibitory effect on excitatory inputs) when the apical trunk expresses LVA $Ca^{2+}$ channels.

### 3.3 Shunting inhibition by $I_h$ current can also occur for spatially distributed stimuli

Until now, we have stimulated the neuron with a single input in the soma or in the dendrite at a time, while it is expected that many excitatory synaptic inputs are needed to induce an AP. For this reason, we were not always able to initiate APs by stimulating distal dendrites without using unrealistically strong inputs (Fig 2C, 2D, 2H and 2I). We thus went on to explore the effects of $I_h$ channel activity on AP induction in L5PCs when the stimuli arrive simultaneously at different locations of the dendrite. To do this, we injected AMPAR- and NMDAR-mediated glutamatergic synaptic currents into randomly picked locations at a given distance from the soma. For each set of synapse locations, we searched for the threshold conductance for inducing a somatic AP. We repeated the procedure $N_{samp}$ = 40 times to obtain distributions of threshold conductances, which we used for statistical tests between $I_h$-blocked and control neurons.

We first distributed the synapses all along the apical (Fig 4A, cyan to blue) or basal (Fig 4A, red to orange) dendrite and activated them simultaneously. In both of these cases, presence of

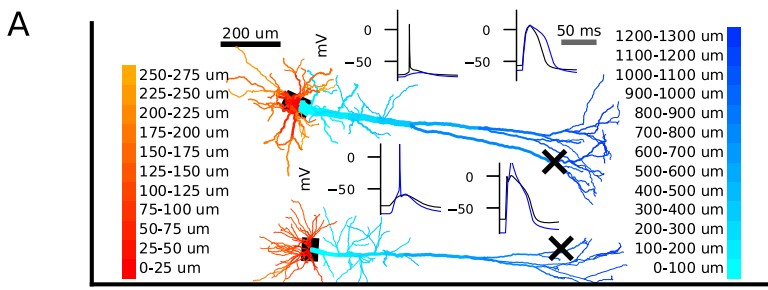

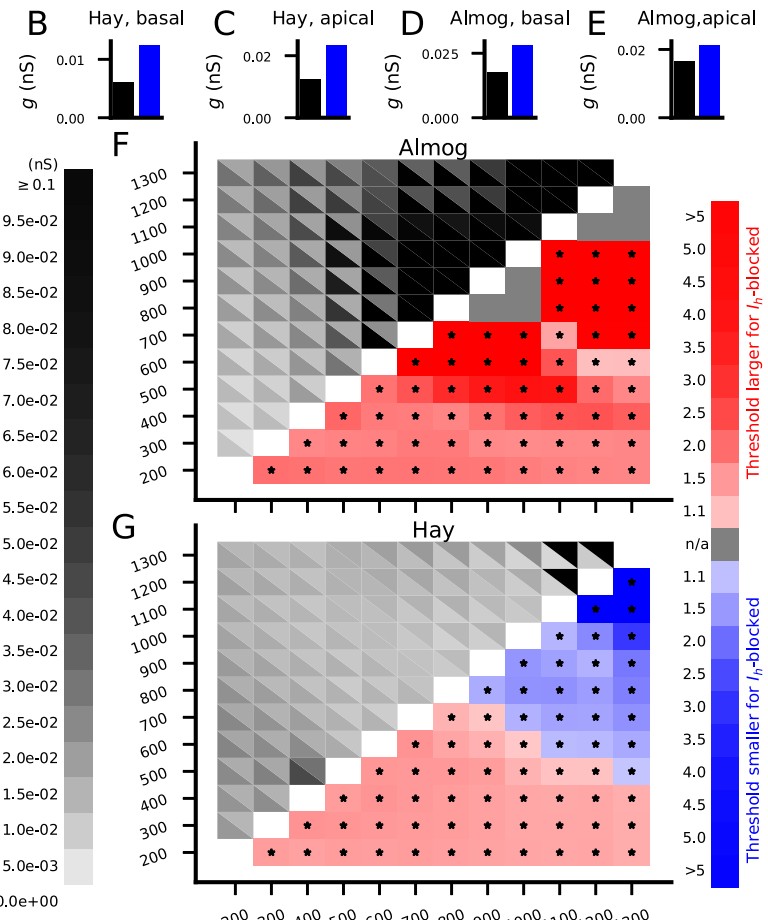

**Fig 4. $I_h$ channels can shunt spatially distributed excitatory synaptic inputs when the inputs arrive at the distal dendrite. A**: Almog- (top) and Hay-model (bottom) morphology color-coded according to distance from soma. Insets: Membrane potentials recorded at soma (left insets) or at a distance 1000 μm from the soma (right insets; exact location of the recording marked with 'x' in the neuron morphology) in response to synaptic stimuli distributed across dendritic locations 800–1200 μm from the soma. Upper inset panels show the Almog model data and the lower panels show the Hay model data; the black curves show the neuron response in presence and the blue curves in the absence of $I_h$ currents. The synaptic conductance amplitude was 10% larger than the minimum of the two threshold conductances (with and without $I_h$ currents) needed for a somatic AP—namely, 0.046 nS in the Almog model (upper insets), and 0.013 nS in the Hay model (lower insets). **B–E**: Threshold conductance for a set of 2000 excitatory synapses (simultaneously activated) to induce an AP in the Almog (B–C) or Hay (D–E) model in a control (black) or $I_h$-blocked (blue) L5PC. In (B) and (D), the synapses were uniformly distributed along the basal dendrite, whereas in (C) and (E), the synapses were uniformly distributed along the apical dendrite. The threshold conductance was always larger in the absence of $I_h$ channels. **F**: *Upper left grid*: The threshold conductances for a set of 2000 excitatory synapses (simultaneously activated) to induce an AP in the Almog model. The synapses were uniformly distributed along the apical dendrite between distances $[x_1,x_2]$ from the soma where the parameters $x_1$ (x-axis) and $x_2$ (y-axis) ranged from 200 μm to 1300 μm in intervals or 100 μm. In each grid slot, the color of the upper right triangle indicates the threshold

conductance in the control neuron whereas that of the lower left triangle indicates the threshold conductance in the $I_h$-blocked neuron. *Lower right grid*: The factor by which the threshold conductance of the $I_h$-blocked neuron is larger (red) or smaller (blue) than that of the control neuron for the stimuli distributed in sections ranging from 200–1200 µm (y-axis) to 300–1300 µm (x-axis). These factors were calculated as fractions of the threshold conductances $g_{\text{Threshold},I_h\ \text{blocked}}/g_{\text{Threshold,control}}$. Asterisks indicate statistically significant differences (U-test, p<0.05/66). Grey squares represent sections where the set of stimuli was unable to induce an AP for all tested conductances (until 0.1 pS) in both $I_h$-blocked and control L5PC. The threshold conductance was always larger in the absence of $I_h$ channels (red squares), indicating a weak shunting effect of the $I_h$ channels or no shunting at all. **G**: The experiment of (F) repeated for the Hay model. The threshold conductance was larger in the absence of $I_h$ channels for proximal inputs (red squares) but smaller for distal inputs (blue squares), indicating a strong shunting effect of the $I_h$ channels in the distal apical dendrite.

$I_h$ currents lowered the AP threshold in both the Almog (Fig 4B and 4C) and Hay (Fig 4D and 4E) models. This was expected, since we observed similar trend in the f-I curves for somatic stimulus (Fig 1E and 1F) and in the threshold currents for proximal inputs at the apical dendrite (Fig 2B and 2G)—and although the distal inputs showed an opposite trend in the Hay model, the proximal inputs are likely to be more determinant for AP initiation than distal inputs of the same strength. We next distributed the synapses on the apical dendrite at intervals $[x_1, x_2]$ from soma where we varied $x_1$ and $x_2$ from 200 to 1300 µm (furthest point of the apical dendrite) in intervals of 100 µm. In the Almog model, $I_h$ activity always facilitated the neuron firing (Fig 4F), while in the Hay model, $I_h$ activity facilitated the neuron firing for proximal inputs and raised the threshold for distal inputs (Fig 4G). Apart from the most distal parts of the Almog-model neuron (Fig 4F), APs could be initiated with physiologically realistic (conductance of a single synapse < 1 nS) stimulation of all parts of the dendritic trees of the model neurons. Despite the variability in threshold conductance depending on the exact location of the synaptic inputs at the given locations, all differences between $I_h$-blocked and control neurons were statistically significant (U-test, p<0.05; Bonferroni corrected by the number (66) of statistical tests), except for the ones where no AP was initiated for any tested synaptic conductance either in the control or $I_h$-blocked neuron (grey squares in the lower right triangle of Fig 4F). This applied also to the Almog model supplemented with densely distributed LVA $Ca^{2+}$ channels to form a hot zone of $Ca^{2+}$ channels in the apical dendrite (S6(A) Fig). Taken together, our results suggest that spatially distributed stimuli with physiologically realistic conductances may be shunted by $I_h$ channels in presence of a hot zone of LVA $Ca^{2+}$ channels, but without the hot zone the $I_h$ channels only contribute to lowering the AP threshold of spatially distributed stimuli.

### 3.4 cAMP-enhancing modulation of distal dendrites and cAMP-inhibiting modulation of proximal dendrites as well as GABAergic inhibition of the basal dendrites strengthen the shunt-inhibitory role of $I_h$ channels

The above analyses highlighted the bimodal effect of $I_h$ currents when the neuron was uniformly affected by $I_h$ blocker. However, pyramidal neurons express a large set of neurotransmitter receptors that are non-uniformly distributed or selectively activated by presynaptic connections. Here, we explored how the shunt/excitation dichotomy of the $I_h$ channels is affected by interaction of different neurotransmitter systems in different parts of the dendritic tree.

First, we replicated the qualitative result of Fig 4G using whole-cell neuromodulation—i.e., we used cAMP-inhibiting modulation of $I_h$ instead of $I_h$ blockage and cAMP-enhancing modulation as an $I_h$-facilitator. As expected, the Hay model predicted that whole-cell cAMP-inhibiting modulation of $I_h$ channels decreased the threshold for distal inputs and increased the threshold for proximal inputs, while cAMP-enhancing modulation had the opposite effect

(S6(B) and S6(C) Fig). The statistical significance of these results remained even if we assumed a weaker, HCN1-like modulation, both when a normal (S7 Fig) and faster (S8(A)–S8(D) Fig) time constant of $I_h$ inactivation was used.

We next quantified the effect of $I_h$ activity on the threshold conductance of apical stimuli in presence of synaptic inputs arriving at the basal dendrite. When glutamatergic stimulation was applied at the same time to both basal and apical dendrite of the Hay-model neuron, $I_h$ current had a mostly excitatory effect except for the stimuli applied to the very distal parts of the apical dendrite (Fig 5A). By contrast, when GABAergic stimulation was applied to the basal dendrite within a window of ±25 ms from the glutamatergic stimulation at the apical dendrite, we observed a stronger shunting inhibition effect of the $I_h$ current (Fig 5B). Representative dendritic membrane potential time courses in response to apical stimulation at distances 800–1200 μm from the soma are displayed in Fig 5C. We repeated these simulations using simpler, current-based analyses (Fig 5D and 5E). Our model of cAMP-inhibiting neuromodulation had similar effects: neuron-wide cAMP-inhibiting modulation in presence of glutamatergic inputs at the basal dendrite increased the AP threshold only for very distal apical stimuli, while in the presence of GABAergic stimulation it increased the AP threshold also for middle-apical (600–700 μm) inputs (S9 Fig). These results suggest that simultaneous activation of excitatory synapses at basal and apical dendrites constrain the shunt-inhibition effect of $I_h$ channels to the most distal parts of the apical dendrite.

Finally, we analyzed the effects of a partial neuromodulation of the apical dendrite and interactions of cAMP-inhibiting and enhancing neuromodulation in affecting the AP threshold in the apical dendrite. When proximal (up to 500 μm from the soma) apical dendrite of the Hay model was under cAMP-enhancing neuromodulation (Fig 6A, blue) or cAMP-inhibiting neuromodulation (Fig 6B, blue), the AP threshold for a set of simultaneously activated glutamatergic synapses was lowered or increased, respectively, across the stimulus locations. In this experiment, we distributed the synapses across dendritic locations at a distance $[x_1, x_1+100\ \mu m]$ from the soma, where we varied $x_1$ from 200 to 1200 μm. When the modulation of the proximal apical dendrite by cAMP-enhancing neuromodulation was accompanied by cAMP-inhibiting modulation of the distal (from 500 μm from the soma onwards) apical dendrite, the neuron was made yet more excitable at the distal dendrites (Fig 6A, magenta)—and accordingly, when the cAMP-inhibiting modulation of the proximal apical dendrite was accompanied by cAMP-enhancing modulation of the distal apical dendrite, the cell was yet less excitable at the distal apical dendrite (Fig 6B, magenta). When we repeated the experiment using wider distributions of synapses as done in the experiments of Figs 4F, 4G and 5A and 5B, a similar result was obtained (Fig 6C and 6D). We also performed a similar experiment using single current-based stimuli along the apical dendrite and different L5PC models. In these experiments, cAMP-enhancing (inhibiting) modulation of proximal apical $I_h$ channels lowered (raised) the AP threshold of a single current-based stimulus, both in the Hay and Almog models and in the Almog model expressing a hot zone of LVA $Ca^{2+}$ channels (S10 Fig, blue). Moreover, this modulation in combination with cAMP-inhibiting (enhancing) modulation of the distal dendrite further lowered (raised) the AP threshold in the distal apical dendrite (S10 Fig, magenta)—this applied to the Hay model (S10(A) and S10(D) Fig) and the Almog model with the hot zone of $Ca^{2+}$ channels (S10(C) and S10(F) Fig), but the native Almog model predicted in-between levels of excitability for neuromodulator combinations (S10(B) and S10(E) Fig). Likewise, the combination of the opposite modulations at proximal and distal dendrite was more effective than the modulation of distal apical dendrite alone (S11 Fig). Taken together, our modelling results suggest that $I_h$ channel-activating neuromodulation in the proximal apical dendrite increases the L5PC excitability, and that in L5PCs expressing a

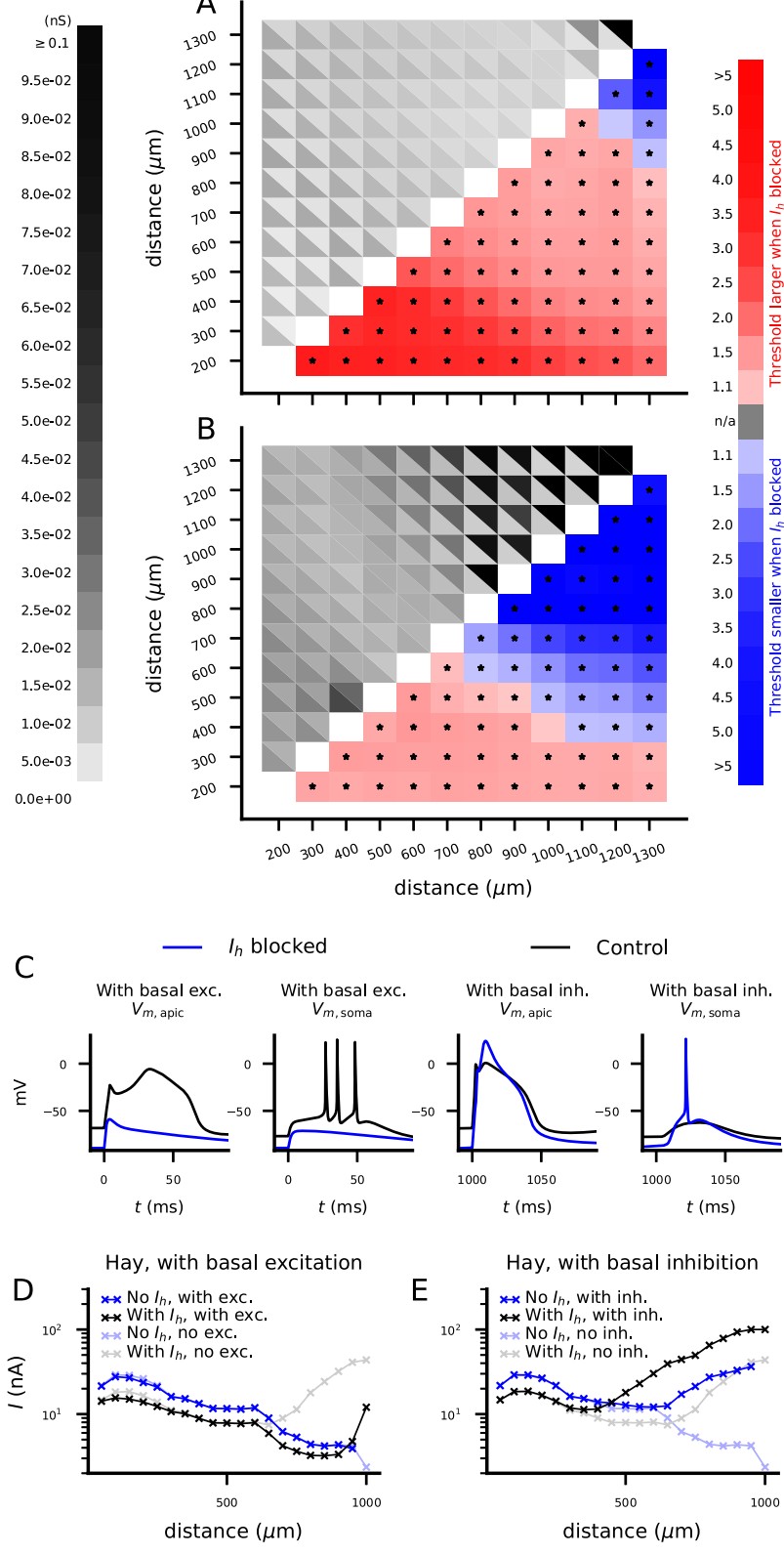

**Fig 5. The shunting effect of $I_h$ is constrained to distal parts of apical dendrite by simultaneous glutamatergic stimulation and strengthened by GABAergic stimulation of the basal dendrite. A–B**: The Hay-model predictions for the AP threshold for spatially distributed apical dendritic stimulation in presence and absence of $I_h$ channels when

the basal dendrite is simultaneously stimulated with glutamatergic (A) or GABAergic (B) inputs. See Fig 4G for details. For the glutamatergic stimulation of the basal dendrite, we distributed 2000 glutamatergic synapses across the basal dendritic tree, and their conductance was set 80% of the AP threshold conductance in the control condition (18 pS, see Fig 4D). For the GABAergic stimulation of the basal dendrite, we distributed 500 GABAergic synapses across the basal dendritic tree, and we randomly picked their activation times from the uniform distribution ±25 ms from the apical activation time and set their conductance 200 pS. As in Fig 4G, the threshold conductance was larger in the absence of $I_h$ channels for proximal inputs (red squares) but smaller for distal inputs (blue squares). This figure shows that the frontier between shunting and excitatory effects of the $I_h$ currents was pushed further or brought closer by simultaneous excitation or inhibition, respectively, at the basal dendrite. **C**: Membrane potentials at a distance 1000 μm from the soma and at the soma in response to a combination of basal and apical glutamatergic stimuli as in (A) (two left-most panels) or a combination of basal GABAergic stimuli and apical glutamatergic stimuli as in (B) (two right-most panels). A single trial shown for apical synapses distributed at distances 800–1200 μm both in presence (black) and absence (blue) of $I_h$ currents. The synaptic conductance amplitude at the apical dendrite was 10% larger than the minimum of the two threshold conductances (with and without $I_h$ currents) needed for a somatic AP—namely, 0.0089 nS when combined with basal excitatory stimuli (two left-most panels) and 0.013 nS when combined with basal inhibitory stimuli (two right-most panels). **D–E**: The AP threshold for a single current-based apical dendritic stimulus in presence (black) and absence (blue) of $I_h$ channels when the basal dendrite is simultaneously stimulated with glutamatergic (D) or GABAergic (E) inputs. The dim curves show the data from Fig 2G where there was no stimulation of the basal dendrite. In (D), we stimulated the basal dendritic compartment at 50 μm from the soma with a conductance-based, alpha-shaped glutamatergic (reversal potential 0 mV) input whose amplitude was 100 nS, and in (E) with a GABAergic (reversal potential -80 mV) input with amplitude 5000 nS.

hot zone of LVA $Ca^{2+}$ channels, the neuron is made yet more excitable by $I_h$-suppressing neuromodulation in the distal apical dendrite.

## 4 Discussion

An intriguing property of $I_h$ channels is that they can exert opposite effects on the excitability of the L5PC. Previous research has explored several explanations, and point to a complex interaction of several factors, such as site of stimulation or input strength. However, experimental work is in general limited in the number of variables that can be manipulated simultaneously. To overcome these limitations, we used biophysically detailed computational modelling. The results from two different single-neuron models converged in an explanation to the differential modulation of $I_h$ on L5PC excitability: the presence of a hot zone of $Ca^{2+}$ channels determines how the site of incoming stimulation will interact with $I_h$ currents. Our study helps explain the great flexibility that these channels confer to the pyramidal neurons. Such a flexibility is crucial for higher order brain functions needed for cognitive processing [41]. Furthermore, a disbalance in the modulatory role that $I_h$ channels has on L5PC may explain abnormal cortical processing in disorders whose symptomatology is hard to account for by simple or linear reductions in excitatory capacity.

Apical and basal dendrites of L5PCs provide distinct input to the neuron: while distal areas receive input from feedback, thalamic, and subcortical sources, proximal areas receive feed-forward inputs. In turn, these sites are themselves compartmentalized. $I_h$ channels are distributed unequally along the apical dendrite, with increasing density towards the apical tuft in the distal extreme, and therefore have a specific role in modulating the inputs that arrive at different points of the dendrite. We first studied how $I_h$ channels can modulate whether and how inputs from these different regions interact with the inputs to the perisomatic region. In the present work, we applied on one hand the broadly used Hay model, which was developed to reproduce a wide range of L5PC features measured in a population of neurons [25], and on the other the Almog model, which was built to closely reproduce the features of a single L5PC using a parameter peeling approach [26]. Both models gave largely consistent findings, which speaks for the robustness of the found effects. By simulating excitatory inputs at increasing distance from the soma, we found that distance of the site of stimulation from the soma determines the effect of $I_h$ current on cell firing: $I_h$ current is excitatory (i.e. it lowers the AP

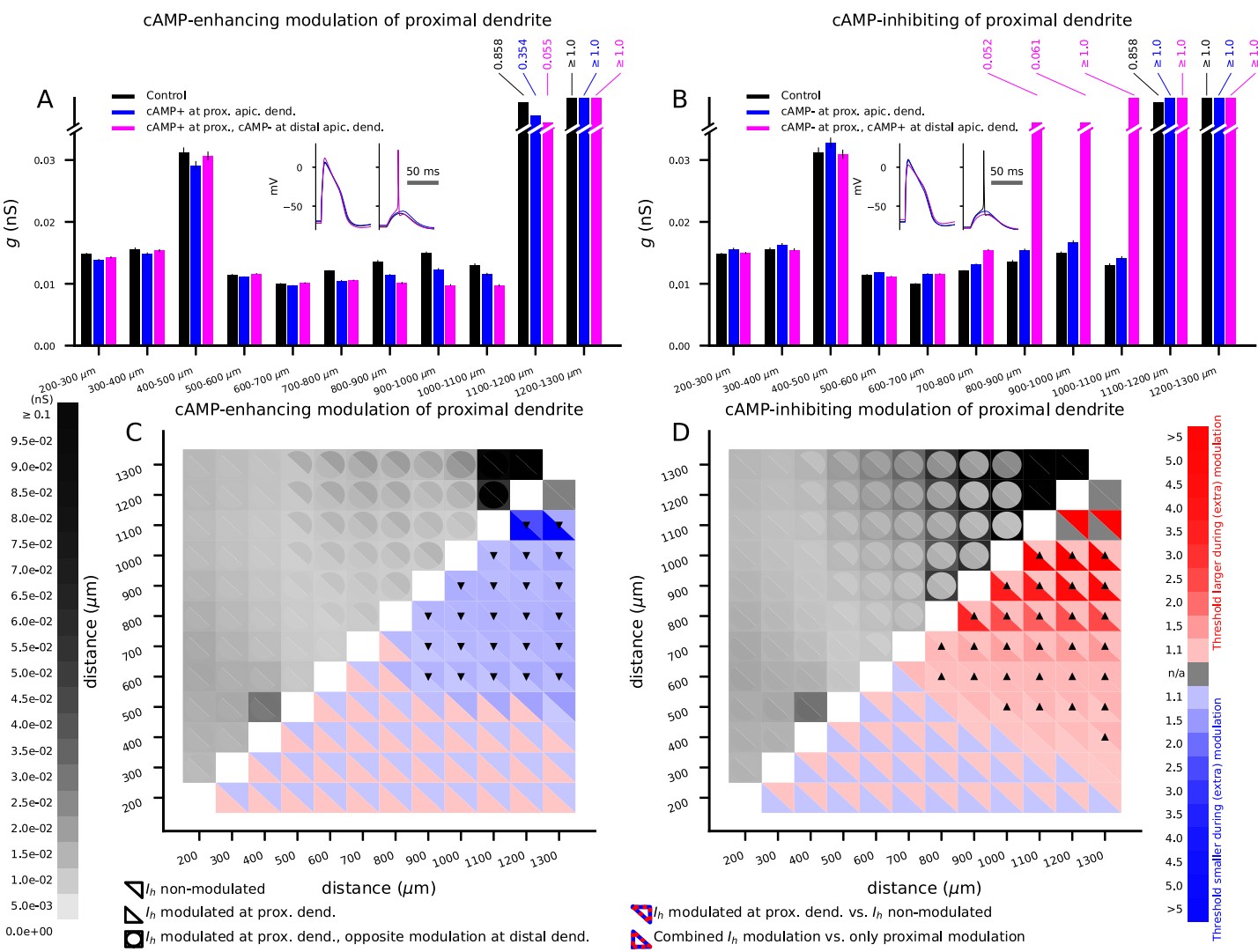

**Fig 6. Combination of cAMP-enhancing and cAMP-inhibiting neuromodulation can increase or decrease the AP threshold throughout the apical dendrite in the Hay-model L5PC. A**: The threshold conductances for a set of 2000 excitatory synapses (distributed across apical dendrite at distances $[x_1, x_1 + 100 \ \mu m]$ from the soma, simultaneously activated) to induce an AP in the Hay model. Black: Control Hay model. Blue: cAMP-enhancing modulation of the $I_h$ channels in proximal apical dendrite (located nearer than 500 μm to the soma). Magenta: cAMP-enhancing modulation of proximal apical dendrite and cAMP-inhibiting modulation of the distal apical dendrite. Insets: Membrane potentials 1000 μm from the soma (left) or at the soma (right) in response to synaptic stimuli distributed across 800–900 μm from the soma. The synaptic conductance amplitude was 10% larger than the minimum of the three threshold conductances (control, proximal apical dendrite under cAMP-enhancing neuromodulation, or combination of cAMP-enhancing and inhibiting neuromodulation at proximal and distal apical dendrites) needed for a somatic AP—namely, 0.011 nS. **B**: The experiment of (A) repeated with the opposite modulation, i.e., cAMP-inhibiting modulation of proximal apical dendrite and cAMP-enhancing modulation of the distal apical dendrite. Insets: same as in (A), but opposite modulation. The amplitude of the synaptic conductances was 0.015 nS. **C**: The experiment of (A) repeated by varying the width of the synapse distribution as in Fig 4F and 4G. *Upper left grid*: In each slot, the color of the upper right triangle indicates the threshold conductance in the control neuron, the lower left triangle indicates the threshold when the proximal apical dendrite was under cAMP-enhancing neuromodulation, and the color of the circular frame surrounding the slot indicates the threshold when the proximal apical dendrite was under cAMP-enhancing and distal apical dendrite under cAMP-inhibiting neuromodulation. *Lower right grid*: The factor by which the threshold conductance of the modulated neuron was larger (red) or smaller (blue) than that of the less modulated neuron. In each slot, the color of the upper right triangle indicates the comparison between control neuron and proximally modulated neuron, and the lower left triangle indicates the comparison between proximally modulated neurons with and without cAMP-inhibiting modulation of the distal apical dendrite. The markers denote series of statistically significant differences (U-test, p<0.05/198): Upward (△) or downward (▽) triangle markers mean that the threshold for proximally modulated neuron was significantly larger or smaller than that of the control neuron and the threshold for proximally (cAMP-enhancing neuromodulation) and distally (cAMP-inhibiting neuromodulation) modulated neuron was larger or smaller than that of the neuron that was only proximally modulated, respectively. **D**: The experiment of (C) repeated with the opposite modulation. The upward triangles of panel (C) largely overlap the downward triangles of panel (D) at the distal parts of the apical dendrite, suggesting that for distal inputs the effects of proximal cAMP-enhancing neuromodulation and distal cAMP-inhibiting neuromodulation are cumulative, and likewise, the effects of proximal cAMP-inhibiting neuromodulation and distal cAMP-enhancing neuromodulation are cumulative. By contrast, the mixed blue and red squares for stimuli reaching proximal dendrites indicate compensatory effects of proximal cAMP-enhancing neuromodulation and distal cAMP-inhibiting neuromodulation, and vice versa.

threshold) if the stimulus is applied to the proximal apical or basal dendrite and inhibitory (i.e. it increases the AP threshold) if the stimulus is applied to the distal apical dendrite. This is consistent with experimental work, where HCN channels were found to have inhibitory effects at the distal apical dendrite, but excitatory effects at the proximal sites [42]. A distance-dependent shunting effect of $I_h$ was also found in CA1 cells [12]. Furthermore, we observed that the reversal point between inhibitory and excitatory effects occurred at a specific distance from the soma, which could not be explained solely by the increase in density of $I_h$ channels with distance to the soma.

We sought to further disentangle the reason behind the effect of inputs' distance to the soma on $I_h$ modulation. We observed that the distance effect in presence of $I_h$ current suffered a reversal around the site corresponding to the hot zone of $Ca^{2+}$ channels. The Hay model including T-type $Ca^{2+}$ channels, and a tuned Almog model where we added a hot zone of $Ca^{2+}$ channels, allowed us to study the effect of removing this hot zone. Our simulations with the Hay model and the tuned Almog model agreed on $I_h$ having a purely excitatory effect in the absence of a hot zone of LVA $Ca^{2+}$ channels in the apical dendrite, while in the presence of the hot zone, $I_h$ currents increased the AP threshold for distal apical dendritic stimuli. Our results are thus in line with [12], where blockage of $I_h$ (both ZD7288 and -/- knockout of HCN1 were experimentally tested and computationally modelled) from CA1 neurons resulted in larger distal dendritic $Ca^{2+}$ events, but unlike the study of [12], our results additionally suggest a clear AP facilitating role for $I_h$ channels in the proximal dendrite regardless of the presence or absence of T-type $Ca^{2+}$ channels. Altogether, our results suggest that $I_h$ activity mostly contributes to higher L5PC excitability but that, in the presence of strong LVA $Ca^{2+}$ channels, $I_h$ channels can also shunt depolarizing inputs at the distal apical dendrite.

## 4.1 Neuromodulation of $I_h$ channels in L5PCs and its effects on excitability

The understanding of L5PC neuromodulation has increased during the past years although much remains to be revealed [43] (note that in this work, by the term neuromodulation we exclusively refer to chemical neuromodulation). Cholinergic M2 receptors are expressed in the axons and basal and apical dendrites of L5PCs [44, 45]. Cortical pyramidal cells of all layers express D1-type receptors (i.e., dopamine receptors that increase cAMP levels upon activation) in their somata and apical dendrites [46]. D2-type receptors were found in a subclass of prefrontal cortical L5PCs projecting to subcortical areas [47]—these receptors are (similar to cholinergic M2 receptors) coupled to Gi/o proteins, which, opposite to Gs proteins, inhibit the cAMP production. As for norepinephrine receptors, cortical pyramidal cells express $\beta_2$-adrenergic receptors in layers II/III as well as V [48] and $\alpha_2$-adrenergic receptors across cortical layers [49]. As for serotonergic receptors, the majority of mPFC pyramidal cells express Gi-coupled 5-HT1 receptors [50], and the Gs-coupled 5-HT7 receptor is strongly expressed in prefrontal cortical pyramidal cells although only early in development [51]. While there is a lack of data on the effects of neuromodulators on the gating properties of many ion channels that regulate L5PC excitability, the voltage-dependence of $I_h$ channels is known to be shifted toward more hyperpolarized potentials (channels more likely closed) by cAMP-inhibiting (i.e., Gi-activating) neuromodulators [23, 24] and toward depolarized potentials (channels more likely open) by cAMP-enhancing (i.e., Gs-activating) neuromodulators [16, 52]. Based on these data, activation of receptors of these neuromodulatory systems were simulated by a shift of the voltage dependence of $I_h$ inactivation by 5–10 mV toward negative (cAMP-inhibiting neuromodulation) or positive (cAMP-enhancing neuromodulation) potentials. We should also highlight the opposing effects that all these four neuromodulators can have on pyramidal cell activity, and $I_h$ channels in particular. Activation of D1 and D2 receptors have opposing effects on

intracellular cAMP concentration, and both have effects of L5PC excitability [3, 53]. Similarly, activation of $\beta_2$-adrenergic receptors that canonically couple with Gs proteins also activate Gi/o proteins [54], and $\alpha_2$-adrenergic receptors are Gi-coupled too. Activation of $\alpha_2$-adrenergic receptors has been observed to increase apical tuft excitability in L5PCs although it slightly hyperpolarizes the resting membrane potential [13], whereas activation of $\beta_1$-adrenergic receptor depolarized the L5PC membrane and increased the neuron's perisomatic excitability [55]—both effects were shown to be mediated by $I_h$ channels. As for cholinergic receptors, cortical pyramidal cells also express M1 receptors [56, 57], which are coupled to Gq proteins, and these receptors also regulate L5PC excitability [58]. Moreover, similar to cholinergic M2 receptors, serotonergic 5-HT1A receptors (Gi/o-coupled) shift the half-inactivation voltage of $I_h$ currents toward hyperpolarized potentials [24], and these receptors mediated an inhibitory net effect in a subgroup of prefrontal cortical L5PCs [59, 60]. By contrast, activation of serotonergic 5-HT7A receptors (Gs-coupled) shift the half-inactivation toward hyperpolarized potentials [20], and this has been shown to contribute to depolarization of L5PCs in young rats [51] and cats [61]. In sum, multiple neuromodulatory systems affect L5PC excitability through receptors that modulate $I_h$ activity through the cAMP intracellular pathway. Here we took advantage of this common mediator to model site-specific effects of neuromodulatory input.

When assessing the effect of cAMP-enhancing modulation, our results showed that in the presence of a hot zone of $Ca^{2+}$ channels, cAMP-enhancing modulation decreased L5PC excitability when stimulated at the distal apical dendrite while cAMP-inhibiting modulation increased it. This is consistent with the proposed role of D1-receptor-mediated effects of dopamine and $\alpha_{2A}$-receptor-mediated effects of noradrenaline in the prefrontal cortex [62]. D1 agonists increase HCN channels' open probability while $\alpha_{2A}$ agonists decrease it. During normal brain processing state, increased $I_h$ channel activity would help reduce the influence of lateral input to superficial layers in order to facilitate the processing of the neuron's preferred stimulus [63] and enhance sustained firing during working memory tasks [62, 64, 65]. Another study on the effect of acetylcholine on dendro-somal integration in L5PCs found that optogenetic stimulus-evoked release of acetylcholine led to specific modulation of the apical dendrite [66]. In particular, when paired to both somatic and dendritic stimulation, it was found to greatly augment its effect in the somatic excitability [66]. While the authors concluded that $I_h$ was not among the most important contributors to the modulation, our model suggests that M2-mediated modulation of $I_h$ works the same way as the M1-mediated effects they observed: it increases the distal apical excitability in the presence of a hot zone of $Ca^{2+}$ channels while it suppresses the excitability in response to somatic stimulation. Our results of AP firing in response to somatic current injections under cAMP-enhancing or cAMP-inhibiting neuromodulation (Fig 1) are also in line with [55] and [67] where activation of $\beta_1$-adrenergic receptor or dopaminergic D1 receptor, respectively, led to increased firing in response to somatic stimulation, and on the other hand with [3] and [59], where activation of dopaminergic D2 receptor or serotonergic 5-HT1A receptor, respectively, led to decreased firing in response to somatic stimulation.

Our computational modelling framework provided an efficient tool for flexibly studying the effects of different scenarios with combined neuromodulation. We collected our central model predictions for L5PCs in Table 3. Our simulations with a combination of cAMP-enhancing neuromodulation of $I_h$ channels in the distal apical dendrite and cAMP-inhibiting neuromodulation in the proximal dendrite yielded stronger shunting of distal apical inputs than either of these modulations alone (and likewise, cAMP-inhibiting modulation in the distal apical dendrite and cAMP-enhancing modulation in the proximal dendrite facilitated the neuron response to distal apical stimuli than either neuromodulation alone; Fig 6). Our results obtained with the two L5PCs models strongly suggest that the $I_h$-mediated shunting of distal

**Table 3. Predictions for how the excitability of L5PCs in response to glutamatergic inputs at proximal and distal apical dendrites is changed by modulation of $I_h$ channels.** Higher excitability (decreased AP threshold) is denoted by ↑, and lower excitability (increased AP threshold) is denoted by ↓. The combinations ↑↑ and ↓↓ represent cases where the combination of cAMP-inhibiting and cAMP-enhancing modulations at different parts of the dendrite caused stronger increase or decrease to the L5PC excitability than any of the two neuromodulations alone.

| | L5PC with a hot zone | L5PC without a hot zone |
|---|---|---|
| cAMP-enhancing modulation | Proximal: ↑, distal: ↓ | Proximal: ↑, distal: ↑ |
| cAMP-inhibiting modulation | Proximal: ↓, distal: ↑ | Proximal: ↓, distal: ↓ |
| cAMP-enhancing modulation at proximal and cAMP-inhibiting modulation at distal | Proximal: ≈ or ↑, distal: ↑↑ | Proximal: ↑, distal: ↑ |
| cAMP-inhibiting modulation at proximal and cAMP-enhancing modulation at distal | Proximal: ≈ or ↓, distal: ↓↓ | Proximal: ↓, distal: ↓ |

inputs requires the hot zone of LVA Ca²⁺ channels: without the hot zone, the $I_h$ neuromodulation of the proximal apical dendrite determines whether the L5PC excitability is increased (cAMP-enhancing modulation) or decreased (cAMP-inhibiting modulation), and the opposite neuromodulation at the distal apical dendrite only has a mildly compensating effect on the L5PC excitability (S10(B) and S10(E) Fig). Moreover, our simulations suggest that the apical dendritic regime where cAMP-enhancing modulation of $I_h$ decreases the L5PC excitability is narrowed down by simultaneous glutamatergic stimulation and expanded by simultaneous GABAergic inhibition of the basal dendrites (Fig 5).

Our results have functional implications, because the proximal and distal parts of the apical dendrite receive neuromodulatory inputs from different sources [68, 69]. In addition, although experimental approaches often study the effect of single neuromodulatory systems, our findings are in line with the view that the combination of several systems, which is the most likely scenario during normal brain processing, may exert synergistic effects on neuron excitability [70]. Fully addressing the question of neuromodulators' actions on $I_h$-channel activity is curbed by the age-dependent and inter-species differences in expression of neuromodulatory receptors. For example, serotonergic 5-HT7 receptors are highly expressed at birth but decreases during postnatal development [71], and 5-HT1 receptors were found to be expressed in the basal dendrites of rat L5PCs [72] while in primates they were found to be expressed mostly in axons [73]. Our work makes specific predictions about the interaction between cAMP-enhancing and cAMP-inhibiting neuromodulators, input region and $I_h$ modulation of neuron excitability, but they will be further extended once more data is gathered on the activity in the neuromodulatory sources and their interaction with each other and the cortex.

## 4.2 Role of LVA Ca²⁺ channels and their interaction with $I_h$ channel modulation in L5PCs

Our results consistently suggest that LVA Ca²⁺ currents are needed for the shunting effect of the $I_h$ channels. LVA Ca²⁺ currents are mediated by T-type Ca²⁺ channels, which are functional as single $\alpha_1$ subunits (encoded by *CACNA1G*, *CACNA1H*, or *CACNA1I*, all of which are mRNA-expressed in mouse L5PCs [74]). The membrane expression of these channels is also strongly modulated by the presence of auxiliary subunits $\alpha_2\delta$, $\beta$, and $\gamma$, but the presence of these subunits does not seem to affect their electrophysiological properties [75]. T-type Ca²⁺ channels were found to be expressed in both soma and dendrites of L5PCs [76], and LVA Ca²⁺ currents were observed in electrophysiological experiments of L5PCs in [77, 78] but not in [79]. Since L-type Ca²⁺ channels, which require auxiliary subunits to be functional, are also expressed and functional in L5PCs, the T-type Ca²⁺-channel $\alpha_1$ subunits may also interact with the auxiliary subunits to increase their membrane expression [75]—however, the

distribution and selectivity of the auxiliary $Ca^{2+}$ channel subunits in L5PCs remains unknown. If future research reveals subclasses of thick-tufted L5PCs possessing different magnitudes of LVA $Ca^{2+}$ currents (or substantially different distributions of LVA $Ca^{2+}$ channels), our results will help to classify these groups into $I_h$-shunting and non-shunting subpopulations.

T-type $Ca^{2+}$ channels are also regulated by neuromodulators. Dopamine, mediated by D1Rs, was found to inhibit T-type $Ca^{2+}$ currents in glomerulosa cells from rat adrenal glands [80], and similarly, acetylcholine inhibited Cav3.3-type LVA $Ca^{2+}$ currents in HEK293 cells, mediated by co-transfected M1Rs [81]. By contrast, mAChR agonist carbachol increased T-type $Ca^{2+}$ currents in [82]. Our results suggest that some of these observed effects could in fact be due to the effects of these neuromodulators on $I_h$ currents, which, by altering the resting membrane potential, have downstream effects on the degree of inactivation of LVA $Ca^{2+}$ channels. For example, even the HEK293 cells, which are widely used as a model system for studying transfected ion channels only, natively express mRNA encoding $I_h$ channel subunits [83]. We thus suggest that experimental settings studying the effects of neuromodulators on T-type $Ca^{2+}$ channels should control the interaction of the $I_h$ channels to prevent mixed effects of the two types of channels. Once data on the effects of neuromodulators on ion-channel activity are obtained for both T-type $Ca^{2+}$ channels as well as other ion channels expressed in the L5PCs, these effects could be involved in our modelling framework to produce predictions of total effects—instead of $I_h$-mediated effects only as studied in our work—of neuromodulation on L5PCs.

## 4.3 Comparison with previous computational studies of $I_h$ function on pyramidal cells

Of the previous computational studies, the approach of [34] was methodologically closest to ours although they used a simplified, generic neuron model. Similar to our work, [34] did not assume a coupling of $I_h$ with another current, yet they showed that the interactions between $I_h$ and two other channels, namely M-type $K^+$ channels and T-type $Ca^{2+}$ channels may crucially change the effects of blockage of $I_h$ on neuron excitability. However, they only analyzed these interactions for responses to somatic stimuli, while here we studied the responses to dendritic stimuli at different locations. In particular, they showed that by increasing the M-type $K^+$ channel conductance the effect of $I_h$ blockade was changed from loss to gain of excitability [34]. In their simulations, the change of T-type $Ca^{2+}$ channel conductance did not show similar switch—however, this is most likely due to the high M-type $K^+$ channel conductance or small range of T-type $Ca^{2+}$ channel conductance chosen for these experiments, since the derivatives of the threshold currents with respect to the T-type $Ca^{2+}$ channel conductance of $I_h$-blocked and unblocked neurons are visibly different (Figure 5f of [34]). Our models, by contrast, predicted that M-type $K^+$ channel conductance had little effect on the threshold currents both in absence and presence of $I_h$ currents (S3 Fig). This question begs for additional research, in particular when studying the effects of cholinergic neuromodulation of $I_h$, since the M-type $K^+$ channels are strongly modulated by acetylcholine [84]. Our models, however, highlight the crucial role of LVA (T-type) $Ca^{2+}$ channels in whether $I_h$ channels promote or hinder L5PC excitability.

In another study on the role of $I_h$ current in CA1 pyramidal neurons, [85] predicted that $I_h$ channels shunt synchronized but not unsynchronized inputs to CA1 pyramidal neurons. [86] suggested that the shunting effect seen in experiments of CA1 pyramidal neurons is caused by an interaction of $I_h$ and M-type $K^+$ channels. However, [37] claimed that the conductance of the M-type $K^+$ channels needed for shunting inhibition would be unrealistically high. Instead, they suggested that $I_h$ current is always coupled to another ionic current (i.e., when $I_h$ is

blocked, the other current is blocked too) that drives the shunt-inhibition effects observed in experiments. [38] examined a range of neocortical L5PC models (including the Hay model), and suggested that a coupling of TASK-like channel with the $I_h$ currents as delineated in [37] provided the best fit to experimental impedance amplitude and phase data. While it is possible that different mechanisms modulate $I_h$ influence on neuron excitability on different regions, our models suggest that T-type $Ca^{2+}$ channels and not M-type $K^+$ channels have a role in the shunting or excitatory effects of $I_h$ currents.

The differences between CA1 pyramidal neurons and L5PCs should also be kept in mind when making comparisons to the studies mentioned above. CA1 pyramidal cells have shorter apical trunks (90 μm in mouse CA1 [87]) than L5PCs (300–400 μm in mouse visual cortex [88]), and the distribution of $I_h$ channels seems to be more nonlinear in neocortical L5PCs than in CA1 pyramidal neurons [2].

## 4.4 Relevance for cognition and behavior

Increasing evidence proposes the ability to integrate information within L5PCs as the mechanism behind some of the most complex processing capabilities of the brain. An important factor is that L5PCs present a clear segregation of inputs (for a review, see [89]). Inputs to the basal dendrite or perisomatic zones are related to the preferred sensory stimulus of the neuron, that is, the information itself that the neuron transmits as part of a feedforward circuit. These inputs can be thus referred to as the specific "content" that a L5PC neuron encodes. On the other hand, inputs to the apical dendrite arrive from higher-order thalamus, feedback loops from prefrontal cortex, subcortical structures such as amygdala, and neuromodulatory regions [90, 91]. The information contained in these inputs encodes arousal or vigilance state, goal information, or state predictions [92]. These apical inputs modulate the strength of the effect that the input to the somatic compartment has on the neuron's excitability, without changing the preference between different stimuli. Thus, the apical inputs act as the "context" that exerts a modulation over the content that arrives to the basal input. For example, high arousal-related neuromodulatory activity enhances the transmission of salient stimuli in the neurons that process these specific stimuli, and thus provide the context that enhances the processing of important information. The ability to integrate context and content is at the core of higher-order brain functions. Brain activation during tasks and behavior is highly dependent on e.g. task goals, emotional or arousal state, and internal states determined by the autonomic system [93, 94]. Our work adds to the existing literature on how these interactions are implemented at the level of single neurons.

Recent theoretical and empirical work suggests the mechanism of "apical amplification" [95] or "dendritic integration" [96] to be at the core of conscious processing. According to these views, only stimuli that are prioritized by contextual information reach the status of being consciously perceived. Furthermore, this cellular mechanism recapitulates the two main features of consciousness, meaning its level and content [97]. Arousal-related neuromodulatory systems like the ones studied here (i.e. noradrenaline, dopamine or acetylcholine) would set the level of consciousness by providing an alertness signal [89]. The arousing or prioritizing effect of neuromodulatory signals (e.g. acetylcholine) is related to downregulation of $I_h$, and was modelled here as a shift in the voltage dependence of $I_h$ current. Consistent with the apical amplification framework, we found an increase in neuron excitability in response to distal apical stimulation following this neuromodulatory effect. However, our simulations with combinations of basal and apical dendritic stimuli predicted that the more there is excitatory drive at the basal dendrite, the smaller the region in the apical dendrite where the $I_h$ current has the shunting effect (Fig 5). Our results thus suggest that the $I_h$ current would more likely

act as a shunt inhibitor in the mode of "apical drive" [98] where the main excitatory drive arrives at the apical dendrite than in the mode of "apical amplification" where both apical and basal dendritic stimulation is needed for an AP.

Our model predictions for the effects of multiple neuromodulatory systems on ability of L5PCs to integrate apical and basal inputs have special relevance in the mental conditions where consciousness is altered. For example, psychotic hallucinations, anesthesia, or dreaming are particular states where the integration of content is disconnected from the context [89, 98, 99]. Such conditions are complex and challenging to describe, because what is altered seems to be the meaning or interpretation of the sensory processing, thus speaking of an impairment in the capacity to contextualize the information. Of the most common mental disorders with hallucinations, altered integration of context-dependent and sensory inputs to L5PCs could be a common pathophysiological feature particularly in schizophrenia, as suggested previously [40, 100–103]. Our findings of the need of both LVA $Ca^{2+}$ channels and $I_h$ currents for the shunting effect of the $I_h$ channels are interesting for schizophrenia research since both $I_h$ and LVA $Ca^{2+}$ channels, alongside serotonergic, dopaminergic, and cholinergic receptors, are products of risk genes of the mental disorder [104–106]. Furthermore, neuromodulatory systems and their interaction likely play a crucial role in the pathophysiology of schizophrenia [107]. Computational work like the one presented here offers a methodological approach to unify the different levels at which this and other mental disorders express their symptoms and phenotypes.

## 4.5 Notes on the magnitude of neuromodulation-mediated voltage shifts of $I_h$ inactivation

The main body of our results was obtained assuming a substantial voltage shift typical to heterotetrameric HCN1/HCN2 channels (half-inactivation voltage shifted by 7–14 mV by increased cAMP [108, 109]) or homomeric HCN2 channels (half-inactivation voltage shifted by 12–17 mV by increased cAMP [16, 110, 111])—we modelled the effects of cAMP-enhancing modulation using a half-inactivation voltage shift of 5–10 mV. However, for homomeric HCN1 channels shifts of 2 mV [112] and 4.3 mV [111] have been observed. It should be noted that although the above data and the data on HCN2-channel modulation [16] were based on measured effects of direct increase of intracellular cAMP, voltage shifts of similar magnitude have been observed when dopamine was applied in the extracellular medium (e.g., 30 μm dopamine shifted the half-inactivation voltage of $I_h$ currents in layer I interneurons by +7 mV in [113]). Both HCN1 and HCN2 are expressed in L5PCs [74, 114], and it is likely—although not explicitly shown—that they are co-localized and form heteromeric $I_h$ channels. Our analysis carried out with a weaker cAMP sensitivity (half-inactivation voltage shifted by ±2–4 mV) confirms that our results on the cAMP-enhancing and cAMP-inhibiting modulation of L5PC excitability hold, although with smaller amplitudes, even if all $I_h$ channels were of the HCN1-type (S7 Fig). Our predictions for the effects of neuromodulation (S8(A)–S8(D) Fig) and blockage (S8(E) Fig) of $I_h$ channels on L5PC excitability were also unaffected if a smaller time constant, typical to homomeric HCN1 channels, was used. In sum, although $I_h$ channel subunit composition affects the magnitude of the effect of neuromodulation, the directions of the effects of neuromodulation, as summarized in Table 3, are not affected by the type of $I_h$ channel.

The shifts of the $I_h$ half-inactivation voltage used in our model may be over- or underestimated also due to the uncertainty about the strength of the activation of different neuromodulatory systems *in vivo* in L5PCs. Most of the experimental work addressing this question *in vitro* used bath application of neuromodulators or agonists of neuromodulatory receptors,

which may cause stronger or weaker cAMP production or inhibition than what happens in endogenous neuromodulation *in vivo*. The neuromodulatory systems considered in this work have been suggested to operate through volume transmission rather than wired transmission [45, 115–117] (note, though, that this view has been challenged in [118]), and if this is the case, it is unsure whether the neuromodulator concentrations in the vicinity of the neuromodulatory receptors are large enough to significantly modulate the $I_h$ channels. It has been estimated that volume-transmitted neuromodulators can locally reach micromolar concentrations. Namely, maximal acetylcholine concentrations of 2–5 µm were measured by amperometry in the PFC [119], and a computational modelling study estimated that extracellular dopamine concentration in the PFC can reach 250–1000 nM upon stimulation of dopaminergic axons [120]. As for noradrenaline and serotonin, their concentrations are expected to reach levels comparable to or higher than that of dopamine since the average levels of these neuromodulators are higher in the cortex [121, 122]. Although these estimates of neuromodulator concentrations are lower than some of the bath concentrations used in the *in vitro* studies of $I_h$ channels, the modulation of $I_h$ has been confirmed with physiologically realistic neuromodulator concentrations as well. For example, a modest dopamine concentration of 500 nM caused a significant D2-mediated decrease in L5PC perisomatic excitability [3]. As for the lower-affinity D1 receptors, small micromolar (5 µm) dopamine concentration was found to alter the lateral pyloric excitability through increased $I_h$ activity [123]. Also a relatively small concentration (10 µm) of serotonin significantly altered a subpopulation of L5PCs in rodent PFC [60]. Importantly, activation of neuromodulatory receptors in L5PCs in response to endogenous neuromodulation and their effects on cellular or network electrophysiology have also been confirmed using antagonists of the underlying receptors. Namely, in [124], D1-receptor antagonism impaired an LTP in layer V of PFC caused by high-frequency stimulation of layer II in presence of picrotoxin, suggesting a non-assisted activation of D1 receptors and their contribution to synaptic plasticity in pyramidal cells. In another study, direct stimulation of the VTA *in vivo* in rats resulted in transitions to up-states in prefrontal cortical L5PCs in a manner indicating monosynaptic effects, and the duration of these up-states was significantly shortened by D1 antagonism [125]. Importantly, we replicated the effects of a physiologically realistic concentration (500 nM) [3] on perisomatic excitability with an $I_h$-current half-inactivation voltage shift of -10 mV using the Hay model (S2(E) and S2(F) Fig). Although more research is needed to map the employed neuromodulator concentrations to the measured intracellular cAMP concentrations and further to $I_h$ voltage-dependence, this result suggests that our range of half-inactivation voltage shifts is likely to be in the correct order of magnitude.

## 4.6 Future directions

In this work, we restricted our analysis on the contributions of $I_h$ currents to single-L5PC excitability. Previous computational modelling studies analyzed the effects of $I_h$ currents on network phenomena such as resonance to oscillations of different frequencies [126–128], and recently the $I_h$ channel dynamics have also been specifically constrained for a computational model of human L5PCs [129]. $I_h$ current of L5PCs, and particularly the inactivation dynamics thereof, were also found important for local field potentials (LFP) in [130, 131]. Our approach of studying the effects of $I_h$ neuromodulation could be directly applied to the analysis of network phenomena as well. In particular, the dependence of working memory-like network activity (as modelled in, e.g., [30]) on neuromodulation of $I_h$ channels could provide important insights into the mechanisms in which neuromodulators shape cognitive processes. Furthermore, modelling of intracellular cAMP concentration and its effects on the $I_h$ activation would

be beneficial since the concentration of cAMP varies across dendrites [132], and thus $I_h$ channels in one part of the dendritic tree may be more prone to negative cAMP modulation and others to positive one. Future models of $I_h$ activity could also benefit from integrating more biochemical interactions between the $I_h$ channels and other voltage-gated channels, in particular $Ca^{2+}$ channels, as done in [133].

## Supporting information

**S1 Fig. $I_h$ channel distribution in the two models. A–B**: Illustration of the $I_h$ channel conductance along the dendritic tree in Almog (A) and Hay (B) models. Black compartments indicate low $I_h$ conductance and green compartments indicate high $I_h$ conductance—see panels (C) and (D) for absolute values. **C–D**: The $I_h$ channel conductance along the apical dendrite with respect to the distance from the soma in Almog (C) and Hay (D) models.
(PDF)

**S2 Fig. $I_h$ activation in the dendrites increases the frequency of action potentials in response to somatic DC in L5PCs. A–B**: The frequency of APs (y-axis) in response to somatic DC of a given amplitude (x-axis) in Almog (A) and Hay (B) model neurons under up- or down-regulated $I_h$ channels. Black: control neuron. Blue: $I_h$ conductance blocked in the apical dendrite. Red: $I_h$ conductance increased by 100% in the apical dendrite. **C–D**: The frequency of APs in response to somatic DC in Almog (C) and Hay (D) model neurons under different neuromodulatory states. Black: control neuron. Blue: cAMP-inhibiting neuromodulation. Red: cAMP-enhancing neuromodulation. **E**: Membrane potential time course data from [3] measured from a control L5PC (black) and an L5PC when bath-applied with 500 nM dopamine (blue). Data digitized from Figure 6A of [3]. **F**: Somatic membrane potential time course predicted by the Hay model for a 0.44-nA somatic stimulation of 1 second in control L5PC (black) and under cAMP-inhibiting neuromodulation (blue). The reversal potential was adapted to -88.4 mV (originally -85 mV in the Hay model) to account for the differences in intracellular $K^+$ concentrations (140 mM in [3], 120 mM in the experiments underlying the Hay model). The shift of -10 mV in the half-inactivation voltage of $I_h$ in the Hay model had the same effect on the number of APs in response to 1-second stimulus (decreased from 7 to 5 APs) as the bath-application of 500 nM dopamine in [3].
(PDF)

**S3 Fig. Blockage of fast $Na^+$ channels abolishes spiking and blockage of LVA $Ca^{2+}$ channels from the Hay model prohibits the shunting of distal apical dendritic stimuli by $I_h$ currents, but blockage of other ion channels does not affect the qualitative behaviour where $I_h$ currents make the neuron more excitable by strong proximal inputs and less excitable by strong distal inputs at the apical dendrite.** See Fig 2 for details. **A**: LVA $Ca^{2+}$ channels blocked. **B**: HVA $Ca^{2+}$ channels blocked. **C**: M-type $K^+$ channels blocked. **D**: Persistent $K^+$ channels blocked. **E**: Transient $K^+$ channels blocked. **F**: Persistent $Na^+$ channels blocked. **G**: $Ca^{2+}$-dependent $K^+$ channels (SK channels) blocked. **H**: Kv3.1-type $K^+$ channels blocked. **I**: Transient $Na^+$ channels blocked. Black curves: the named ion channel blocked. Blue curves: the named ion channel and the $I_h$ channel blocked.
(PDF)

**S4 Fig. A single-compartment model of a distal apical dendritic section of the Hay-model L5PC predicts that the shunting effect of $I_h$ currents is mediated by a high degree of LVA $Ca^{2+}$ channel inactivation in the resting state in presence of $I_h$ currents. A–C**: Time courses of the membrane potential (A) and the activation (B) and inactivation (C) variables $m$ and $h$ (Eqs 3 and 4) of the LVA $Ca^{2+}$ channels in the control condition when the

compartment was stimulated with an alpha-shaped synaptic conductance of 2 nS. **D–F**: Time courses of the membrane potential (D) and the activation (E) and inactivation (F) variables $m$ and $h$ when $I_h$ channels were blocked. Note the significantly larger values of $h$ in (F) compared to control (C). **G**: Time course of the membrane potential of a model compartment, where the LVA $Ca^{2+}$ current is replaced by an artificial LVA current species where the values of the activation variable $m$ (Eq 3) are directly taken from $I_h$-blocked simulation (E) and those of the inactivation variable $h$ (Eq 4) are taken from the control simulation (C). This model compartment produces a *milder* response than either the control (A) or $I_h$-blocked neuron (D). **H**: Time course of the membrane potential of a model compartment, where the LVA $Ca^{2+}$ current is replaced by an artificial LVA current species where activation variable $m$ is taken from the control simulation (B) and the inactivation variable $h$ is taken from the $I_h$-blocked simulation (F). This model compartment produces a *stronger* response than either the control (A) or $I_h$-blocked neuron (D). **I**: The membrane potential time courses from panels (A), (D), and (G–H) overlaid. The observation that the response in (H) reached (and went beyond) that of the $I_h$ blocked neuron indicates that increase in dendritic spike magnitude caused by $I_h$ blockage is due to altered levels of inactivation, not activation, of LVA $Ca^{2+}$ channels.
(PDF)

**S5 Fig. The Almog model with a hot zone of LVA $Ca^{2+}$ channels produces a $Ca^{2+}$ spike in response to strong apical dendritic stimulation. A–B**: Time courses of the membrane potential of the original Almog model (A) and the Almog model with a hot zone (B) in response to short (0.2 ms) apical dendritic square-pulse current at 800 μm from the soma, recorded at the location of input (green) and at soma (black). The stimulus intensity was varied from 5.0 nA (left) to 100 nA (right). The Almog model with a hot zone of LVA $Ca^{2+}$ produces APs for smaller stimulus intensity and exhibits a stronger dendritic spike that leads to a burst of somatic APs instead of a single spike.
(PDF)

**S6 Fig. $I_h$ current-mediated shunting of distal apical dendritic stimuli shown by different simulations. A**: Predictions of the Almog model with a hot zone of $Ca^{2+}$ channels. The upper left grid shows the threshold conductances for a set of 2000 excitatory synapses to induce an AP, and the lower right grid shows the factor by which the threshold conductance of the $I_h$-blocked neuron is larger (red) or smaller (blue) than that of the control neuron. See Fig 4F for details. **B–C**: Predictions of the Hay model for the cAMP-enhancing (B) or cAMP-inhibiting (C) neuromodulation compared to the non-modulated neuron. *Upper left grid*: The threshold conductances for a set of 2000 excitatory synapses to induce an AP in the Hay model. In each grid slot, the color of the upper right triangle indicates the threshold conductance in the control neuron whereas that of the lower left triangle indicates the threshold conductance in the neuron under cAMP-enhancing (B) or cAMP-inhibiting (C) neuromodulation. *Lower right grid*: The factor by which the threshold conductance of the neuron under cAMP-enhancing (B) or cAMP-inhibiting (C) neuromodulation is larger (red) or smaller (blue) than that of the control neuron.
(PDF)

**S7 Fig. Effects of weak $I_h$ channel modulation on apical dendritic excitability in Almog and Hay model. A–B**: Predictions of the Almog model with a hot zone of $Ca^{2+}$ channels for the threshold currents in control neuron and neuron with $I_h$ channels weakly modulated by cAMP-enhancing (+2 mV; A) or cAMP-inhibiting (-2 mV; B) neuromodulation. **C–D**: Predictions of the Hay model for the threshold currents in control neuron and neuron with $I_h$

channels weakly modulated by cAMP-enhancing (+4 mV; C) or cAMP-inhibiting (-4 mV; D) neuromodulation. See Fig 4F for details.
(PDF)

**S8 Fig. A–D**: Effects of weak $I_h$ channel modulation on apical dendritic excitability in Almog and Hay model when a faster $I_h$ inactivation was assumed. The experiments of S7 Fig were repeated using 4 times smaller time constants of $I_h$ inactivation ($\tau_\infty$) than in the original Hay and Almog models. See S7 Fig for details. **E**: Threshold current amplitudes for 2-ms square-pulse inputs at the apical dendrite at different distances from the soma using 4 times smaller time constants of $I_h$ inactivation. Black: Hay-model control neuron, blue: Hay-model neuron with $I_h$ blockage. The amplitudes are very similar to those with the original time constants (Fig 2G).
(PDF)

**S9 Fig. The shunting effect of cAMP-enhancing neuromodulation and the excitability-increasing effect of cAMP-inhibiting neuromodulation neuromodulation are further constrained to the distal apical dendrite by glutamatergic stimulation of the basal dendrite and expanded by GABAergic stimulation of the basal dendrite. A–D**: The four panels show the Hay-model predictions for the effects of cAMP-enhancing (A,C) or cAMP-inhibiting (B, D) neuromodulation on the AP thresholds of apical dendritic stimulation in the presence of simultaneous glutamatergic (A–B) or GABAergic (C–D) stimulation of the basal dendrite. *Upper left grids*: The threshold conductances for a set of 2000 excitatory synapses to induce an AP in the Hay model. In each grid slot, the color of the upper right triangle indicates the threshold conductance in the control neuron whereas that of the lower left triangle indicates the threshold conductance in the neuron under cAMP-enhancing (A–B) or cAMP-inhibiting (C–D) neuromodulation. *Lower right grids*: The factor by which the threshold conductance of the neuron under cAMP-enhancing (A,C) or cAMP-inhibiting (B,D) neuromodulation is larger (red) or smaller (blue) than that of the control neuron.
(PDF)

**S10 Fig. Combination of cAMP-enhancing and cAMP-inhibiting neuromodulation can increase or decrease the AP threshold throughout the apical dendrite in L5PCs expressing a hot zone of LVA $Ca^{2+}$ channels. A–C**: Threshold amplitude for a 2-ms current input applied to the apical dendrite at a given distance (x-axis) from soma according to Hay model (A), Almog model (B), or Almog model with a hot zone of LVA $Ca^{2+}$ channels (C). Black: control neuron. Blue: neuron with proximal apical dendrite under cAMP-enhancing neuromodulation. Magenta: neuron with proximal apical dendrite under cAMP-enhancing neuromodulation and distal apical dendrite under cAMP-inhibiting neuromodulation. **D–F**: The experiment of (A)–(C) repeated with opposite modulation, i.e., control neuron (black) and neuron with cAMP-inhibiting modulation of proximal apical dendrite with (magenta) or without (blue) cAMP-enhancing modulation of the distal apical dendrite.
(PDF)

**S11 Fig. Combination of cAMP-enhancing and cAMP-inhibiting neuromodulation can increase or decrease the AP threshold throughout the apical dendrite in the Hay-model L5PC.** The experiment of Fig 6 was repeated such that the one-sided neuromodulation (proximal dendrite modulated and distal dendrite unmodulated, i.e., the blue data of Fig 6) was replaced by the alternative one-sided neuromodulation (distal dendrite modulated, proximal dendrite unmodulated, green data). See Fig 6 for details.
(PDF)

**S12 Fig.  A**: Membrane potential (upper panels) and $Ca^{2+}$-channel reversal potential (lower panels) time series in response to short (0.2 ms) supra-threshold current stimuli at the apical dendrite at distances 200 (left)—1000 (right) μ according to the Almog model. The stimulus amplitude was 30 nA, except at the distance of 800 μ and 1000 μ amplitudes 100 and 300 nA, respectively, were used. **B**: Membrane potential (upper panels) and $Ca^{2+}$-channel reversal potential (lower panels) time series in response to short (0.2 ms) supra-threshold current stimuli at the apical dendrite at distances 200 (left)—1000 (right) μ according to the Hay model. The stimulus amplitude was 30 nA, except at the distance of 1000 μ amplitude 100 nA was used. **C**: The experiments of Fig 1G were repeated using neuromodulatory voltage shifts of ±10 mV.
(PDF)

## Acknowledgments

We thank William A. Phillips for his comments on the manuscript. UNINETT Sigma2 resources (project NN9529K), and resources from CSC, IT Center for Science, Finland (project 2003397) were used for simulations.

## Author Contributions

**Conceptualization:** Tuomo Mäki-Marttunen, Verónica Mäki-Marttunen.

**Data curation:** Tuomo Mäki-Marttunen.

**Investigation:** Tuomo Mäki-Marttunen.

**Methodology:** Tuomo Mäki-Marttunen.

**Supervision:** Verónica Mäki-Marttunen.

**Writing – original draft:** Tuomo Mäki-Marttunen, Verónica Mäki-Marttunen.

**Writing – review & editing:** Tuomo Mäki-Marttunen.

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
