## [Decision Letter · Decision Letter 0]

25 Apr 2022

Dear Mäki-Marttunen,

Thank you very much for submitting your manuscript "Excitatory and inhibitory effects of HCN channel modulation on excitability of layer V pyramidal cells" for consideration at PLOS Computational Biology.

As with all papers reviewed by the journal, your manuscript was reviewed by members of the editorial board and by several independent reviewers. In light of the reviews (below this email), we would like to invite the resubmission of a significantly-revised version that takes into account the reviewers' comments.

We cannot make any decision about publication until we have seen the revised manuscript and your response to the reviewers' comments. Your revised manuscript is also likely to be sent to reviewers for further evaluation.

Sincerely,

Joanna Jędrzejewska-Szmek, Ph.D.

Associate Editor

PLOS Computational Biology

Daniele Marinazzo

Deputy Editor

PLOS Computational Biology

Reviewer's Responses to Questions

**Comments to the Authors:**

Reviewer #1: 

In this study, the authors used two computational models of layer V pyramidal cell (L5PC) and examined the impact of hyperpolarization-activated cyclic nucleotide-gated (HCN) channel modulation on the spike activity. They found that dense expression of voltage-dependent calcium channels is necessary to evoke shunting inhibition of synaptic inputs on the distal apical dendrite. In addition, the authors simulated the impacts of the neuromodulator on the efficiency of synaptic transmission in the pyramidal cell by modulating the voltage dependency of the HCN channels.

The involvement of high-density calcium channels in the shunt effect is only addressable through computational simulation. The impacts of neuromodulators can be partly tested empirically (for example, local perfusions of agonists and photo-uncaging of neurotransmitters), but such experiments require a lot of time and experience. Therefore, the authors' computational approaches in this study are reasonable.

The main findings of this study provide a potential mechanism to control the synaptic transmission in L5PC in both healthy and pathophysiological conditions. There are some concerns about the model of HCN channels in this study. Therefore, this study should only be accepted after resolving these concerns.

<major concerns="">

1. Models of HCN channel and the effects of neuromodulators

The authors refer to two experimental studies on which their implementation of voltage dependency of HCN channel conductance critically depends (Zhao et al., 2016 and Byczkowicz et al., 2019). These studies examined the impacts of acetylcholine (ACh) and dopamine on the striatal neuron and cerebellar mossy fiber boutons, both of which primarily expressing HCN2 channels.

However, the authors of this study examined the behavior of the cortical neuron. There is co-expression of HCN1 and HCN2 channels in pyramidal cells. The discrepancy between relevant HCN subtypes raises a major concern.

HCN1 channels and HCN2 channels have significantly different activation speeds and sensitivities to cyclic AMP (He et al., 2014, PMID: 24184323). When co-expressed in the same cell, these properties differ depending on the fraction of subtypes (Chen et al., 2001, PMID: 11331358). Therefore, the degree of modulation used in this study’s computational model is not realistic and potentially far greater than in vivo conditions. This has critical implications for the veracity of their computational model because one of their major findings depends on the shifted voltage dependency.

The authors must clarify the fraction of HCN1 and HCN2 channels in L5PC and accordingly control the HCN channel properties in their model. If they find similar results as to what was reported here, then it’s interesting.

2. Concentrations of ACh and dopamine

In the referenced studies, the modulations of voltage dependency were performed by either controlling intracellular cAMP through a glass pipette during the patch-clamp recording or by the perfusion of the antagonist in the extracellular solution. The concentration of neuromodulators reported by the submitting authors is likely beyond the physiological operating range of cortical neurons reported previously.

For example, Hironaka et al. (2001) reported that the ACh concentration in the rat prefrontal cortex ranged from 10 nM to 30 nM (PMID: 11368961). On the other hand, the dose of ACh used in the referenced work (Zhao et al., 2016) used 50 μM of ACh. Additionally, Spühler and Hauri (2013) developed a simulation allowing in silico estimation of dopamine concentrations in prefrontal cortex (PMID: 23951205). Spühler and Hauri found that the concentration of dopamine in the prefrontal cortex likely falls between 9.5 nM to 250 nM in the steady state and 23.4 nM to 989 nM with 26 Hz phasic dopamine release. The submitting authors instead refer to Byczkowicz et al. (2019), which used 200 μM of dopamine in their study.

Zhao et al. (2016) and Byczkowicz et al. (2019) used doses of Ach and Dopamine, respectively, that are more than 200 times high than what was previously reported as possible in vivo. These doses may represent a non-physiological perturbation that are not ideal for a simulation that attempts to reconstruct normal, physiological spiking activity.

The submitting authors need to adjust the effects of neuromodulators based on physiological concentrations. If they find similar results as to what was reported here, then it’s interesting.

<minor concerns="">

1. Authors may need to explain the model of voltage-dependent calcium channels in the methods section because they modulate the parameters of these channels in the results 3.2.

2. There is no explanation for the scale bar in figure 2 A, E, and F.

3. Authors need to explain what the activation variables and the inactivation variables of the low-voltage activated calcium (LVA) channels are in the results 3.2.

4. Authors need to explain what observation in supplementary figure 3 indicates the disappearance of the shunting in the results 3.2.

5. How did authors calculate the factor by which the threshold conductance of the Ih-blocked is changed in figure 4-6? Authors need to mention this in the figure legend or methods.

<optional suggestions="">

The authors only report firing rate changes when evaluating the effectiveness of HCN conductance and neuromodulators. This approach is reasonable because spike activity is the output from neurons. This study could be more informative and attract more interest if the authors provide detailed electrophysiological data (for example, the magnitude of stimulation evoked postsynaptic potentials and the resting membrane potentials). The authors would be able to easily collect these parameters because the NEURON simulator has current-clamp and voltage-clamp functions.

Reviewer #2: There are some very interesting points made in this paper about the distance dependent effects of Ih and about the interplay between Ih and Ca channels. This should be focus of the paper and there should be more information supporting what is known about Ca channel types and locations. All figures would benefit from addition of voltage traces.

Reviewer #3: * Major Comments

** The authors describe interesting differences in the effect of HCN channels depending

on the presence or absence of Ca2+ channle hot zones, but don't mention whether or not

their simulations support the existence of hot zones, or if they predict some L5PCs to

have hot zones and other don't, or if the Ca2+ channels in the hot zones may be themselves

blocked or modulated. It is interesting to note that in Almog & Korgreen (2014), they tried

adding a hot zone and couldn't achieve dendritic spikes. Does your altered version of the

Almog model with the Ca2+ channel hot-zone produce dendritic spikes?

ubiquity of Ca2+ channel hot-zones?

** The effects of HCN blockade, overexpression, and modulation are very small in the Hay model

compared to the Almog model (Fig. 1). The difference between the scales of the F-I curves and

bar graphs for the two models also, at a glance, serves to amplify the difference in the Hay model

relative to the Almog model. I would suggest either placing them on the same y-scale or

mentioning the differences in scale in the figure caption.

It would be very helpful to list either the change or percent change in threshold current

amplitudes for inducing an AP and/or the distance of the half-max values of the F-I curves in the

text to make this difference between the models more clear.

** The difference in effects of HCN blockade, overexpression, and modulation also

begs the question: How different are the distributions of HCN

channels between these models, and how different are the models of Ih used in the two models

themselves. The original papers where the Ih models were developed (Kole, Hallermann,

and Stuart, J. Neurosci. 2006 and Williams and Stuart J. Neurophysiol 83:3177, 2000) should

be cited and a description of the differences between them included in the text.

Furthermore, a more detailed description or supplementary figure showing

the spatial distribution of Ih in these models (like plot of gbar for Ih vs. distance from soma)

might help in explaining this difference. Not only might the differences in HCN models and

distributions explain the differences between the models seen in Fig. 1, they might also

contribute to the effects of the Ca2+ hot zones described later on.

I noticed that the Ih models from each model neuron are copied into the other's folder

(e.g. modulhcn/modeulhcn_almog/IhHay.mod), but I'm curious if you ever tried swapping these

Ih models in the neurons and running even just the F-I curve protocol.

** The authors need to distinguish the neurons they are studying, thick tufted L5PCs with high HCN

channel expression that project to subcortical structures like pyramidal-tract neurons in motor cortex,

to thin tufted L5PCs which have low HCN channel expression and project elsewhere in cortex

(intratelencephalic).

* Minor comments

** The figure captions do a thorough job describing the contents of the figures but do

little to explain the results. Especially for the more complicated figures (Figs. 4-6),

some interpretation of the important results in the figure caption would help with readability.

** Figure three is hard to follow (partly for the same reasons as above). Recommend

including "dim" curves in the figure legend (like control and block Ih) or use

different colors/linestyles; otherwise I needed to keep going back and forth between

the figure, the caption, a previouf figure, and the legend,

** [page 6] The phrase "increased I h activation by lamotrigine application" makes it sound like

lamotrigine is an HCN channel agonist. Please clarify.

** The descriptions of the synaptic stimulation protocols should be outlined in the Methods section;

whereas now they are really only described in detail in the figure legends.

** [page 18] It's worth noting that one of the "range of neocortical L5PC models" examined in Kelley et al. (2021)

was the Hay model, which did not fit the experimental phase data, but that probably does not have

much influence on the phenomena the authors describe. Furthermore, "our models

suggest that when using a realistically morphological model of an L5PC" implies that the models used in

Migliore & Migliore (2012) and Kelley et al. (2021) were not morphologially realistic or detailed, which

they were.

** [page 18] "Thus, the apical inputs act as the context that exerts a modulation over the content that arrives

to the basal input". Please carlify how apical inputs represent "context" while basal inputs represent

"content".

**Have the authors made all data and (if applicable) computational code underlying the findings in their manuscript fully available?**

Reviewer #1: Yes

Reviewer #2: Yes

Reviewer #3: Yes

PLOS authors have the option to publish the peer review history of their article (what does this mean?). If published, this will include your full peer review and any attached files.

Reviewer #1: No

Reviewer #2: No

Reviewer #3: **Yes: **Craig Kelley

Figure Files:

Data Requirements:

Reproducibility:

</optional></minor></major></summary>

---

## [Decision Letter · Decision Letter 1]

27 Jun 2022

Dear Mäki-Marttunen,

Thank you very much for submitting your manuscript "Excitatory and inhibitory effects of HCN channel modulation on excitability of layer V pyramidal cells" for consideration at PLOS Computational Biology.

As with all papers reviewed by the journal, your manuscript was reviewed by members of the editorial board and by several independent reviewers. In light of the reviews (below this email), we would like to invite the resubmission of a significantly-revised version that takes into account the reviewers' comments.

We cannot make any decision about publication until we have seen the revised manuscript and your response to the reviewers' comments. Please address the comments concerning ACh and serotonin concentrations. Your revised manuscript is also likely to be sent to reviewers for further evaluation.

Sincerely,

Joanna Jędrzejewska-Szmek, Ph.D.

Associate Editor

PLOS Computational Biology

Daniele Marinazzo

Deputy Editor

PLOS Computational Biology

Reviewer's Responses to Questions

**Comments to the Authors:**

Reviewer #1: The authors addressed all minor comments. However, major comments I initially expressed were not addressed enough. I uploaded detailed comments regarding the authors' responses as an attachment.

Reviewer #3: The authors satisfactorily addressed my previous concerns.

**Have the authors made all data and (if applicable) computational code underlying the findings in their manuscript fully available?**

Reviewer #1: Yes

Reviewer #3: Yes

PLOS authors have the option to publish the peer review history of their article (what does this mean?). If published, this will include your full peer review and any attached files.

Reviewer #1: No

Reviewer #3: No
---

## [Decision Letter · Decision Letter 2]

15 Aug 2022

Dear Mäki-Marttunen,

Thank you very much for submitting your manuscript "Excitatory and inhibitory effects of HCN channel modulation on excitability of layer V pyramidal cells" for consideration at PLOS Computational Biology. As with all papers reviewed by the journal, your manuscript was reviewed by members of the editorial board and by several independent reviewers. The reviewers appreciated the attention to an important topic. Based on the reviews, we are likely to accept this manuscript for publication, providing that you modify the manuscript according to the review recommendations.

Sincerely,

Joanna Jędrzejewska-Szmek, Ph.D.

Associate Editor

PLOS Computational Biology

Daniele Marinazzo

Deputy Editor

PLOS Computational Biology

[LINK]

Reviewer's Responses to Questions

**Comments to the Authors:**

Reviewer #1: The authors addressed most of my concerns, however I have two minor comments.

The authors pointed out that the density of the dopamine release site in their referenced study (Spühler and Hauri, 2013) is underestimated because Vitrac et al. 2014 reported a higher density than the reference. However, what Vitrac reported in their study is not the dopamine release site, but rather the mean length of the dopaminergic fiber per brain tissue volume (meter / mm^3). This is different from the density of the dopamine release site. The dopamine release site density was reported in the study (Table 2 in Descarries et al., 1987, PMID: 3627435) utilizing the same method as the newly added reference for the noradrenergic terminal density in the revised manuscript (Audet et al. 1998). Unless the authors specifically want to use the medial part of the frontal cortex where the dopamine release site density exceeds 1,000,000/mm^3, the release site density of the dorsal frontal cortex is 60,000/mm^3 (values from Table 2 in Descarries et al., 1987). Therefore, it does not seem appropriate to conclude that the dopamine release site density reported by Spühler and Hauri at 200,000/mm^3 is underestimated.

The comparison of the effect of dopamine on the firing pattern recorded in the brain tissue (Gulledge and Jaffe, 1998) to the effect of the cAMP-modulation in the authors’ study is a reasonable approach to bridging the results between electrophysiological recording and theoretical studies. I think a quantitative description of the similarity between the two studies (ex, input resistance or the number of spikes before and after the neuromodulation) is more objective and reliable than the authors' description in this manuscript (“was in good agreement with…” in p.19).

Reviewer #3: The authors have addressed all my concerns.

**Have the authors made all data and (if applicable) computational code underlying the findings in their manuscript fully available?**

Reviewer #1: Yes

Reviewer #3: Yes

PLOS authors have the option to publish the peer review history of their article (what does this mean?). If published, this will include your full peer review and any attached files.

Reviewer #1: No

Reviewer #3: No

Figure Files:

Data Requirements:

Reproducibility:

References:

---

## [Editor Report · Decision Letter 3]

19 Aug 2022

Dear Mäki-Marttunen,

We are pleased to inform you that your manuscript 'Excitatory and inhibitory effects of HCN channel modulation on excitability of layer V pyramidal cells' has been provisionally accepted for publication in PLOS Computational Biology.

Best regards,

Joanna Jędrzejewska-Szmek, Ph.D.

Academic Editor

PLOS Computational Biology

Daniele Marinazzo

Section Editor

PLOS Computational Biology

---

## [Editor Report · Acceptance letter]

8 Sep 2022

PCOMPBIOL-D-22-00503R3 

Excitatory and inhibitory effects of HCN channel modulation on excitability of layer V pyramidal cells

Dear Dr Mäki-Marttunen,

I am pleased to inform you that your manuscript has been formally accepted for publication in PLOS Computational Biology. Your manuscript is now with our production department and you will be notified of the publication date in due course.

With kind regards,

Anita Estes
